# The Fly Homologue of MFSD11 Is Possibly Linked to Nutrient Homeostasis and Has a Potential Role in Locomotion: A First Characterization of the Atypical Solute Carrier CG18549 in Drosophila Melanogaster

**DOI:** 10.3390/insects12111024

**Published:** 2021-11-13

**Authors:** Mikaela M. Ceder, Frida A. Lindberg, Emelie Perland, Michael J. Williams, Robert Fredriksson

**Affiliations:** 1Molecular Neuropharmacology, Department of Pharmaceutical Biosciences, Uppsala University, 751 24 Uppsala, Sweden; frida.lindberg@farmbio.uu.se (F.A.L.); Emelie.perland@hotmail.com (E.P.); robert.fredriksson@farmbio.uu.se (R.F.); 2Functional Pharmacology, Department of Neuroscience, Uppsala University, 751 24 Uppsala, Sweden; michael.williams@neuro.uu.se

**Keywords:** solute carrier, MFS, MFSD11, CG18549, drosophila melanogaster, locomotion

## Abstract

**Simple Summary:**

The body is dependent on nutrients and ions to work normally. Within the hu-man body there is a group of proteins named transporters or solute carriers. These transporters are vital for the transport of nutrients such as glucose, amino acids, and fats, as well as ions such as sodium, calcium, and potassium. Despite being vital for normal physiology, as well as pathophysiology, a large number (approximately one third) of the transporters are orphans, where information about their expression and function is missing. Here, we aimed to begin to unravel the expression and function of one of these orphan transporters, MFSD11, by studying its orthologue in fruit flies (CG18549). We found that the fly orthologue is expressed in the brain of fruit flies and that it is possibly involved in metabolism and/or locomotion of the flies. The exact mechanism behind the observed behaviors is not fully understood, but our study provides new insights into the expression and function of CG18549. Clearly, these results, among others about the orphan transporters, provide a strong example as to why it is vital to fully characterize them and through that gain knowledge about the body during normal condition and disease.

**Abstract:**

Cellular transport and function are dependent on substrate influx and efflux of various compounds. In humans, the largest superfamily of transporters is the SoLute Carriers (SLCs). Many transporters are orphans and little to nothing is known about their expression and/or function, yet they have been assigned to a cluster called atypical SLCs. One of these atypical SLCs is MFSD11. Here we present a first in-depth characterization of the MFSD11, CG18549. By gene expression and behavior analysis on ubiquitous and brain-specific knockdown flies. *CG18549* knockdown flies were found to have altered adipokinetic hormone and adipokinteic hormone receptor expression as well as reduced vesicular monoamine transporter expression; to exhibit an altered locomotor behavior, and to have an altered reaction to stress stimuli. Furthermore, the gene expression of *CG18549* in the brain was visualized and abundant expression in both the larvae and adult brain was observed, a result that is coherent with the FlyAtlas Anatomy microarray. The exact mechanism behind the observed behaviors is not fully understood, but this study provides new insights into the expression and function of *CG18549*. Clearly, these results provide a strong example as to why it is vital to fully characterize orphan transporters and through that gain knowledge about the body during normal condition and disease.

## 1. Introduction

Transporters are essential for substrate absorption, distribution, metabolism, and elimination, and they are evolutionary conserved in prokaryotes and eukaryotes [1,2,3,4]. The largest protein family clan of functionally related transporter and transporter-like proteins across several phyla, with over 1,000,000 sequences members divided into 87 recognized families [5], is the Major Facilitator Superfamily (MFS) [6,7,8,9]. In the late eighties, all members of the MFS were presumed to facilitate transport of various sugars [10], but today they are known as a diverse transporter family [7,11]. In humans, the largest superfamily of transporters is the SoLute Carrier (SLC) proteins, and approximately one third of the human SLCs populates the MFS Pfam clan ([12], slc.bioparadigms.org, accessed on 16 March 2021). The main function of SLCs is to transport solutes across the plasma and organelle membranes and provide cells with, e.g., carbohydrates, amino acids, lipids, and ions [2,3]. The function of SLCs can roughly be divided into four categories: cotransporters (symporters), exchangers (antiporters), facilitators (uniporters) and orphans (slc.bioparadigms.org, accessed on 16 March 2021) [2,3]. However, lately, two subfamilies of the SLCs, the Proton-assisted Amino acid Transporter (PAT or SLC36) and the System A and System N sodium-coupled neutral amino acid transporter family (SNATs, SLC38), have gained attention for their involvement in not only transport but also nutrient sensing. Moreover, both families have been linked to mammalian target of rapamycin complex 1 (mTORC1) activation and function [13]. Hence, it is not surprising that one fourth of all SLCs are associated with diseases [14]. However, despite the associations with, e.g., neurological diseases and metabolic disorders, only a few SLCs are used as drug targets [15]. Also, one third of the SLCs are still categorized as orphans and, unfortunately, the SLCs remain understudied compared to other integral membrane proteins [12,14].

In 2017, Perland and colleagues suggested that there are more transporter and transporter-related proteins that ought to be classified as SLCs, and they named this group “Atypical SoLute Carriers” [12,16]. A couple of years later, this was found to be true when the SLC superfamily grew from 52 subfamilies with approximately 393 members to 65 subfamilies with 430 members (no pseudo genes counted) (slc.bioparadigms.org, accessed on 16 March 2021). A large proportion of these atypical SLCs have similarities with the MFS and are therefore called Major Facilitator Superfamily domain (MFSD) proteins (MFSD1-MFSD14). Among the 65 subfamilies of SLCs, six subfamilies contain one or more MFSD protein (slc.bioparadigms.org, accessed on 16 March 2021).

Herein, we have studied the orphan MFSD protein MFSD11 (Major Facilitator Superfamily containing Domain 11). MFSD11 is evolutionary conserved, with orthologs found in simple species as *C. elegans*, *C. intestinalis* and *D. melanogaster*, as well as in more complex species as humans, apes, and rodents [17]. Little is known about its function, but genome-wide association studies predict that MFSD11 could be involved in Retinal arteriolar caliber (study accession: GCST002071) [18] and total paired-helical-filament (PHF) tau (study accession: GCST010340) [19]. At present, there are only four publications focusing directly on MFSD11: a characterization study focusing on the expression pattern of MFSD11 in mouse [17]; a broad study about putative SLCs and their connections to sugar concentrations [20]; a study focusing in mutations associated with trastuzumab-resistance [21] and a genomic study where it was suggested to be a novel intellectual disability (ID) candidate [22]. The gene expression of *Mfsd11* in mice is widespread, with expression in both the peripheral and the central nervous system (CNS). The MFSD11 protein is abundantly found in the mouse brain, both in embryonal and adult tissue and it is expressed in neurons positive for both anti-glutaminase and vesicular inhibitory amino acid transporter (VIAAT) [17]. Furthermore, databases as GTEX reports that human MFSD11 is moderately expressed across tissues and its lowest expression is in brain, pancreas and whole blood (Data Source: GTEx Analysis Release V8 (gtexportal.org (accessed on 16 March 2021), dbGaP Accession phs000424.v8.p2, 21 June 2021). There have been speculations that MFSD11 could be involved in metabolism or at least that it is a gene that reacts highly to the nutritional status of the body, i.e., the gene expression of *Mfsd11* has been found to be affected by sugar concentrations and starvation in both *in vitro* and *in vivo* models [17,20].

The aim of this project was to characterize MFSD11 with focus on both expression and function by studying the orthologous gene in *D. melanogaster*, *CG18549*. Expression of the gene was investigated using quantitative polymerase chain reactions (qPCRs) and visualized using an enhancer-trap line that drives green fluorescent protein in *CG18549* positive cells. Starvation and behaviors linked to locomotion, age and stress were studied with the Drosophila Activity Monitor System (DAMS) on ubiquitous and brain specific knockdown flies. We found that (I) *CG18549* expression is regulated by nutrient availability, (II) *CG18549* knockdown flies exhibit an altered locomotor behavior and (III) have an altered stress response. Unfortunately, the exact mechanism behind the observed behaviors is not fully understood. Taken together, the data suggest that *CG18549* has a potential role in locomotion behavior and possibly in metabolism, either by directly or indirectly affecting genes linked to these processes.

## 2. Materials and Methods

### 2.1. Prediction of Secondary and Tertiary Structure

Models of the human MFSD11 and *D. melanogaster* CG18549 were predicted using Phyre2 [23]. The MFSD11 (Uniprot ID O43934) and the CG18549 (Flybase ID FBpp0082101) structures were modeled against the template with the highest confidence, sequence coverage and identity. The protein unc-93 homolog b1 from *mus musculus* (PDB id c7c77B) [24] were used for prediction of MFSD11 and CG18549 secondary and tertiary structures. MFSD11 was modeled with 11% identity, 100% confidence and 94% amino acid coverage (423 out of 449 residues). CG18549 was modeled with 12% identity, 100% confidence and 97% amino acid coverage (424 out of 436 residues). Phyre2 provided backbone images of both the secondary and tertiary predictions. The gradient of the tertiary structures displays helices from N-terminal (dark blue) to C-terminal (red), and unconnected loops in the predictions are parts of the structure that have uncertainties within the model; therefore, they cannot be fully modeled.

### 2.2. Fly Stocks and Maintenance

The w[*]; P{w[+mC]=GAL4-elav.L}3, P{GawB}elavC155 w*;P{FRT (whs)}G13P{tubP-GAL80}LL2, y[1] sc[*] v[1]; P{y[+t7.7] v[+t1.8]=TRiP.HMS01385}attP2 (CG18549 RNAi line 1, Stock no. 39341, tested for functionality with da-GAL4), w[1118] and Pin/Pin; UAS-mCD8-GFP (derived from yw;Pin/Cyo, UAS-mCD8-GFP, stock no. 5130) received from the Bloomington Stock Center, the y[*] w[*]; P{w[+mW.hs]=GawB} CG18549[NP5325]/TM6, P{w[–]=UAS-lacZ.UW23-1}UW23-1 was received from Kyoto Drosophila Stock Center, and P{KK102196}VIE-260B (CG18549 RNAi line 2, Stock no. v107272, tested for functionality with da-GAL4) was received from Vienna Drosophila RNAi Center (VDRC). Da-Gal4 flies were a generous gift from Professor D. Nässel and CSORC flies were a generous gift from Associate Professor and Dr. M.J. Williams (cross of CantonS and OregonR-C, [25]). All flies were crossed into the same w^1118^ background. In all assays, the GAL4 drivers and UAS transgenic flies were crossed to w^1118^ flies and their F1 progeny were used as controls.

All flies, unless otherwise stated, were maintained on enriched Jazz mix standard fly food (Fisher Scientific, Stockholm, Sweden) supplemented with yeast extract (VWR). Fly stocks were maintained at 25 °C in an incubator at 60% humidity on a 12 h:12 h light:dark cycle. Flies crossed to GAL4 drivers and controls were raised at 18 °C until adults emerged. Once collected, adults were raised at either 25 or 29 °C, depending on the driver line, for the appropriate times.

### 2.3. Gene Expression Analysis Using Fluorescent Protein

The gene expression of *CG18549* in the brain of third instar larvae and adult flies were studied by using F1 progenies from the CG18549-GAL4 and Pin/Pin-UAS-GFP cross, and the progenies from each line separate were used as background controls. Larvae were collected and the brain and ventral nerve cord were dissected in 1xPBS-tx on a silicone coated petri dish. The tissues were rinsed three times to remove membranes and excessive artifacts, before being mounted on Superfrost Plus slides in VectaShield (H-1000, Vector Laboratories, Stockholm, Sweden). Images were captured using fluorescent Axioplan 2 imaging microscope. Adult flies were anesthetized with CO_2_ before decapitation and fixation in 8% formaldehyde for one hour, followed by washing in 1xPBS 15 min three times. The brains were retrieved and either mounted on Superfrost Plus slides in VectaShield (H-1000, Vector Laboratories) and imaged using fluorescent Axioplan 2 imaging microscope or mounted on 1% agarose (Conda) in petri dishes for imaging in a Zeiss LSM710 2 Photon microscope with the Zen black software. Experimental and control images were captured with the same exposure time. 

### 2.4. Diet and Starvation mRNA Panels

#### 2.4.1. Diet Panel

This method was modified from [20,26]. All CSORC flies (n = 5, with 20 male flies in each replicate) were five days old and fed for five days, and then euthanized by freezing at −80 °C. The flies were decapitated, and total RNA was extracted from the heads. Data from fly body was used with permission and modified from [20].

The following sugar and yeast (protein) ratios were used when preparing the diets: 10:10 g/dL (control), 2.5:2.5 g/dL, 40:40 g/dL, 2.5:40 g/dL, 10:40 g/dL, 40:2.5 g/dL and 40:10 g/dL.

#### 2.4.2. Starvation Panel

This method was modified from [20]. Briefly, five- to seven-days-old CSORC males (n = 5–7, with 20 flies on each replicate) were collected at 0 (n = 7), 3 (n = 7), 6 (n = 5), 9 (n = 5), 12 (n = 6) and 24 (n = 6) h of starvation before euthanized by freezing at −80 °C. The flies were decapitated, and total RNA was extracted from the heads. Data from fly bodies were used with permission and modified from [20].

### 2.5. RNA Extraction, cDNA Synthesis, Primer Design and qPCR

RNA extraction, cDNA synthesis, primer design and qPCR were performed as described in [20,27]. 

RNA extraction: In total ten flies per replicate were collected for RNA extraction of whole fly, while 20 flies per replicate were collected for RNA extraction of heads. The RNA pellet was dissolved in 20 µL of DEPC water, and the concentration was measured in a ND-1000 spectrophotometer (NanoDrop Technologies, Wilmington, DC, USA).

cDNA synthesis: Two µg of cDNA was synthesized with High-Capacity RNA-to-cDNA kit (Applied Biosystems, Waltham, MA, USA) according to the manufacturer’s instructions. The samples were diluted to 10 ng/µL RNA in sterile water. 

Primer design: Primers were designed with Beacon Design 8 (Premier Biosoft, Palo Alto, CA, USA). Primers are summarized in Table 1, where reference genes are in red and target genes are in black.

qPCR: Ct Values were obtained using the CFX Maestro (BioRad, Stockholm, Sweden) and primer efficiencies were calculated via LinRegPCR software. The delta Ct methods for either one (*Actin42A*) or several reference genes (*Actin42A*, *Rpl11*, *Rp49* and *Rpl13A*) (according to [28]) were used to normalize and calculate the relative mRNA expression presented as fold differences. The log2 fold change (±95% CI) was plotted, and the gene expressions were compared to the Driver control. Expression differences were analyzed with Kruskal–Wallis with Dunn’s comparison or Mann–Whitney (specified in figure legends) using GraphPad Prism, version 5. 

### 2.6. Offspring Counting

F1 progenies from *da-GAL4 > w^1118^* (Driver ctrl), *w^1118^ > CG18549 RNAi* (RNAi ctrl) and *da-GAL4 > CG18549 RNAi* (*CG18549* knockdown) were collected and examined. Both *CG18549* RNAi lines were used, and the counting was performed on progenies from three crosses of each genotype. The mean (±95% CI) are plotted for male and female progenies separated as well as merged for each RNA line. Differences were analyzed using Kruskal–Wallis with Dunn’s comparison. Progenies were investigated for developmental flaws using a Leica M125 microscope with a ProgRes C14 plus camera (Jenoptik) and the ProgRes CapturePro 2.8 Jenoptik Optical system.

### 2.7. RNA Sequencing

Sequencing libraries were prepared from 1 μg total RNA using the TruSeq stranded mRNA library preparation kit (Cat# 20020594/5, Illumina, Uppsala, Sweden) including polyA selection. The library preparation was performed according to the manufacturers’ protocol. Sequencing was performed on one lane on the NovaSeq SP flowcell, paired-end 150 bp read length, using v1 chemistry on a NovaSeq 6000 system (Illumina).

The sample files are stored in the SRA repertoire and can be found under PRJNA689052. The reads were mapped against the D. melanogaster genome assembly 6.09 using STAR mapper (25). Mapping was undertaken against a genome index generated with the FlyBase GTF annotating file for the 6.09 genome assembly to direct mapping toward annotated genes. A total of 326,643,821 transcripts were mapped uniquely and used in subsequent analysis. The assembled transcripts were used in DESeq2 (26) to obtain a final transcriptome assembly and to calculate the relative and differential expression, summarized in Appendix A. Genes were considered to be differentially expressed if they had a log2 fold change >1 or <−1 in the DESeq2 analysis. An ANOVA with FDR correction was used to identify expression differences between all three sample groups, and Wald test with Benjamini and Hochberg multiple correction method was used as post hoc test. The log2 fold changed genes from the qPCR runs were compared with the RNA sequencing log2 fold change and summarized in Table 2.

### 2.8. Drosophila Activity Monitor System

Starvation resistance assay and locomotion, as well as age-dependent locomotion, were performed as described in [29,30]. Details about the stress assay and the filming of start and stop movement are described below. All experiments were performed on five- to seven-days-old male flies except for the age-dependent locomotion where the flies were aged to 13 days and to 21 days. A minimum of eight (range n = 8–30) individual flies were used for each setup; the exact number and driver lines used for the different experiments are described below under the specific subheading and/or specified in figure legends. All data were retrieved via Trikinetic software. Parameters of flies that deceased during the locomotion assays were removed as outliers. Other outliers were removed using GRUBBS outlier test, with α = 0.05. GraphPad Prism v.5 software was used to generate graphs and perform statistical analyses. Total beam breaks, activity (beam breaks) per hour and data points from filming are presented as scatter plots (mean ±95% CI) and total activity was also plotted over time in a point-connected line graph (mean). Starvation resistance was presented as a scatter plot (mean ±95% CI) to give an indication of total hours survived, also a survival proportion plot and a point-connection line graph were constructed to illustrate the survival over time and locomotion. Differences were calculated using Kruskal–Wallis with Dunn’s comparison and/or Mann–Whitney as a post hoc test to calculate exact p-values (specified in figure legends). 

Starvation resistance: In short, adult male flies from Driver control, RNAi control and CG18549 knockdown (*da-GAL4* driver, n = 30) were transferred to 5 mm glass tubes, prepared with 1% agarose, and contained in the DAMS until the last beam break. The last beam crossing per fly was defined as the timepoint of death.

Locomotion: F1 progenies from Driver control, RNAi control and *CG18549* knockdown (da-GAL4 driver n = 30, elav-GAL4 driver n = 10, elav-GAL4:GAL80 driver (onset 1-day post-eclosion) n = 10) were transferred to 5 mm glass tubes, prepared with enriched Jazz mix standard food, contained in the DAMS for 24 to 72 h to record the activity level (locomotion, beam breaks). Age-dependent locomotion was performed on 13 (Driver ctrl n = 9, RNAi ctrl n = 12, *CG18549* knockdown n = 24) and 21 (n = 30) days old male Driver control (elav-GAL4 driver), RNAi control and *CG18549* knockdown flies with the same setup as the other locomotion assays. 

Stress: Stress assay was performed on Driver control (elav-GAL4 driver), RNAi control and *CG18549* knockdown flies (n = 10) for two days (~48 h). Stress was induced by exposing the flies to bright light (LED, 1500 lumens) for 10 min three times (9.00, 12.00 and 15.00/16.00) per day. The total activity over 48 h was plotted in a line graph and the arrows indicate the timepoints where the flies were exposed to light. The total beam breaks for each day were calculated as well as the total beam break one hour after each stressor. 

Filming: A 24 well tissue culture plate was coated with 1% agarose (conda). Adult flies (elav-GAL4:GAL80 driver, n = 8 per group) were transferred to the plate one fly to each well. The flies were left for 5 min before filming started. The film was recorded by a Panasonic HC-V700 Full HD video camera. Flies were filmed for 20 min and the time of activity was manually measured for each fly using a stopwatch.

### 2.9. Effect Size Calculations

Effect sizes were calculated using mean differences (Mean_target_ − Mean_control_) via www.estimationstats.com (accessed on 16 March 2021) according to [31]. Briefly, the mean differences for comparisons are plotted in a Cumming estimation plot. The raw data is plotted on the upper axes; each mean difference is plotted on the lower axes as a bootstrap sampling distribution. Mean differences are depicted as dots; 95% confidence intervals (CI) are indicated by the ends of the vertical error bars, and is bias-corrected and accelerated. Permutation tests are performed with 5000 reshuffles of the control and test labels, and the *p*-values equals the likelihoods of observing the effect sizes.

## 3. Results 

### 3.1. MFSD11 and CG18549 Have Similar Secondary and Tertiary Structure

MFSD11 is known to be evolutionary conserved and is found in several other species. Phylogenetic analyses and amino acid alignments have been performed previously, revealing that the closest homologue to MFSD11 is CG18549 and that they have been reported to have 41.5% identical amino acid residues when performing a global pair-wise alignment [17,27]. Herein, we wanted to study the secondary and tertiary structures to look at similarities in the 2D and 3D structures, which could tell us more about their relationship and how well the results can be translational. The secondary and tertiary structure were predicted using Phyre2 [23]. The homology modeling predicted that MFSD11 and CG18549 have a similar secondary structure as other MFS proteins with 12 transmembrane helices [6], Figure 1A,B. Both proteins were also found to have the MFS loop between the sixth and seventh transmembrane helices, something that have been reported previously for atypical SLCs of MFS type [32,33], Figure 1A,B. Similar to one of their closest relative proteins, UNC93A [32], both MFSD11 and CG18549 were modeled with short N-terminus and C-terminus. Both MFSD11 and CG18549 were modeled against the crystal structure of UNC93B1, and are therefore predicted to have large similarities in the global protein folding patterns, Figure 1C,D. Both predications were modeled with high confidence and amino acid coverage, which indicates that MFSD11 and CG18549 most likely adopts to the overall fold illustrated in Figure 1.

### 3.2. The CG18549 Gene Is Abundantly Expressed in the Larvae and Adult Brain

MFSD11 has previously been reported to be an abundantly expressed gene and protein in mice, and immunostaining was reported to be present in embryos and adult mice [17] with high expression observed in the central nervous system (CNS). Furthermore, through the FlyAtlas [34] a majority of the genes in *D. melanogaster* have been mapped via microarray data and RNA sequencing data [35], and the data is summarized on flybase.org (accessed on 16 March 2021). According to these expression data sets, *CG18549* is abundantly expressed, with moderate expression in the brain and CNS of larvae and adult flies, and high expressions in the Malpighian tubules, fat body and salivary glands [34]. Interestingly, it is also reported by Fly-FISH (a database of *Drosophila* embryo and larvae mRNA localization patterns) that *CG18549* is expressed during different embryo stages [36,37]. However, these data are based on microarray and/or in situ and the expression has so far not been imaged in adult flies. Herein, we focused on the brain expression of *CG18549* in adult flies to be able to compare with previous findings of *CG18549* and *Mfsd11*, which have been based on brain tissue. To be able to visualize the *CG18549* gene expression in brain cells, we crossed the enhancer-trap line *CG18549-GAL4* and a *UAS-GFP* line, which results in progenies that will have *CG18549* positive cells labeled with green fluorescent protein (GFP). Brains were dissected from five- to seven-days-old male flies, and images were captured with a fluorescent microscope, Figure 2. *CG18549-GFP* signals were visible in the brain and ventral nerve cord of third instar larvae (n = 2), Figure 2A,B, but not in the separate lines, Figure 2C,D. GFP was also observed in the adult brain, Figure 2E, and no specific signals were visual in the controls, Figure 2F,G. Unfortunately, a fluorescent Axioplan 2 imaging microscope was not able to image individual cells; therefore, three additional brains were dissected and imaged with a Z-stack using a Zeiss LSM710 2 Photon microscope. The images, Figure 2H–J, revealed that *CG18549* was expressed in different regions of the adult brain and both larger and smaller cells were GFP-positive, and projections were found to be labelled with GFP. However, no double staining was performed and hence, the exact cell type cannot be determined.

### 3.3. Functional Validation of Two RNAi Lines

So far, limited data has been gathered about MFSD11 and its orthologues; to date, no functional characterization has been performed to understand its physiological role. To be able to perform functional studies on *CG18549* and to study its physiological role, one commonly used technique within the research field of transporters is to manipulate the gene expression level. In *D. melanogaster*, there are sophisticated tools and fly lines available to perform knockdown studies on the desired gene. Herein, we decided to initiate the first in-depth functional characterization of *CG18549* by performing RNAi knockdown. Two different *CG18549* RNAi lines were commercially available and used here. They were tested for functionality using the ubiquitous da-GAL4 driver and measure the relative mRNA expression in three crosses: Driver control (*da-GAL4* > *w1118*), RNAi control (*w1118* > *CG18549 RNAi*) and *CG18549* knockdown (*da-GAL4* > *CG18549 RNAi*). The effect sizes of both lines were estimated by calculating the unpaired mean differences between the log2 fold values (Driver control vs. RNAi control, Driver control vs. *CG18549* knockdown, RNAi control vs. *CG18549* knockdown) and two-sided permutation *t*-tests were performed; the log2 fold differences and mean differences are illustrated in Figure 3A, and the statistical results when comparing the means are presented in Figure 3B. Both RNAi lines were confirmed to produce F1 progenies with reduced *CG18549* expression, Figure 3A,B. However, for *CG18549* RNAi line 1, *CG18549* expression was reduced to 50% compared with both controls, Figure 3A, while for the second *CG18549* RNAi line (*CG18549* RNAi line 2) *CG18549* expression was measured to be reduced in both the RNAi control (90% reduction) and the *CG18549* knockdown (95% reduction) flies, Figure 3A,B. Therefore, the second *CG18549* RNAi line was not primarily used during the functional characterization.

### 3.4. No Difference of F1 Progenies from RNAi Controls and CG18549 Knockdown Flies

*CG18549* is reported to be expressed during several embryo stages [36,37], but still its function remains unknown. Therefore, we hypothesized that it is a possibility that a knockdown of *CG18549* could affect development and/or survival of the embryos. Hence, we wanted to study if there were any visible changes in appearance due to changes in developmental cues and/or alterations in the number of progenies that eclosed. The number of eclosed progenies were calculated for each experimental cross and the progenies were examined under a microscope. No differences were observed in appearance of the knockdown flies and the control flies (data not shown) and, in general, no significant differences were observed between the knockdowns and controls, or between the crosses, Figure 3C. However, the *CG18549* RNAi line 1 was found to have a higher number of eclosed progenies compared with the Driver control and *CG18549* knockdown flies, but the statistical power was not high enough, and the results could therefore not be statistically verified, but the trend was observed both in males and females.

### 3.5. CG18549 Expression Is Affected by the Concentration of Sugar and Protein in the Food

Several of the MFSD proteins are known to be involved in the transport of important metabolites such as glucose, amino acids, and fats [12,16]. Furthermore, earlier studies on *CG18549* [20], as well as the mouse orthologue *Mfsd11* [17], reported that the gene expression is affected by nutrient availability. So far, the findings in *D. melanogaster* were observed on samples obtained from the whole-body of the fly, while the data from mice were measured in brain sections. Here, we wanted to focus on the expression in the heads from flies to be able to compare the results from the brains in mice. We hypothesized that we would observe gene alterations as a response to different diets similar to those reported earlier. To test this, the relative mRNA expression of *CG18549* was measured in five- to seven-days-old wildtype (CSORC) flies that were fed diets with different ratios of sugar (S) and protein (Y), Figure 4. *CG18549* was indeed found to be altered in the head tissue. The expression was increased in flies fed a 40:40 g sugar:yeast (S:Y) diet and a 10:40 g S:Y diet, which is similar to what have been reported in mice fed a high-fat diet [17]; while the gene expression was reduced in flies fed 2.5:40 g S:Y and 40:10 g S:Y, Figure 4A. The reduction in flies fed 40:10 g S:Y corresponds to what has been previously reported in flies [20]. However, no difference was measured in flies fed 2.5:2.5 g S:Y, which do not correspond to the reported difference in *CG18549* expression observed in whole-body tissues of flies [20]. Together these results strongly suggest that nutrient concentrations such as sugar (sucrose) and protein (yeast) have a biological effect on the expression of *CG18549*.

Earlier findings in body focused on the sugar content in food; hence, some information regarding gene expression alterations as a response to fluctuating sugar and protein ratios is missing. The changes observed for both tissues vary, most likely due to the difference in how the data is presented. Therefore, we wanted to convert the data collected from the whole-body of flies [20] to log2 fold to be able to make a better comparison between the fly tissues from the body and head, Figure 4B. No difference between the control diet, that is the 10:10 g S:Y, was found, but interestingly *CG18549* was highly altered with a large difference of the means of the samples in the whole-body tissue of flies fed 2.5:2.5 g S:Y diets and no differences were observed in the head tissue sample of flies fed 2.5:2.5 g S:Y. Furthermore, a large difference between the *CG18549* mRNA expression means of the whole-body tissue samples and the head tissue samples obtained from flies fed the 40:10 g S:Y diet was detected, but still these samples were regulated in similar patterns compared with the controls for each tissue, Figure 4B. These results conclude that there are similarities in how the *CG18549* expression is altered in whole-body tissue and head tissue in flies subjected to a high sugar diet and normal diet. However, the results also strongly suggest that the body *CG18549* is more affected after a calorie deficient diet compared with the heads where the *CG18549* expression instead is highly affected by a sugar rich diet, which could be due to the difference in nutrient need and storage within the body compared to the head (brain). 

### 3.6. Starvation Does Not Affect the CG18549 in the Heads of Wildtype Flies

To further elaborate on the involvement of *CG18549* in metabolic pathways and cues, alterations in gene expression in flies subjected to complete starvation was investigated. Previous findings have shown that *CG18549* is affected by starvation and that the expression normalized upon refeeding [20]. However, once again the gene expression was measured in whole-body tissue samples, and since a difference in gene alterations was observed between whole-body tissue and head tissue, we wanted to investigate if it was also true in flies subjected to starvation. To do so, the gene expression of *CG18549* was measured in five- to seven-days-old flies subjected to complete starvation. No differences were observed between the timepoints (0, 3, 6, 9, 12 and 24 h of starvation), Figure 4C.

In the body, upregulation of *CG18549* has been reported after 3 h of starvation [20]. In the heads, one can observe a trend towards upregulation, but no statistical significance was reported. Once again, the data from the whole-body tissue were not presented in log2 fold change and therefore the data from these samples were collected and recalculated to be able to perform a better comparison to the head tissue samples. After the conversion to log2 fold change, the differences were still present in the samples obtained from the whole-body. Interestingly, the log2 fold expression in whole-body tissue and head tissue were found to follow a similar pattern and the means of the control and the 3 h starved flies were similar, Figure 4D. Taken together, these results suggest that *CG18549* is not as affected by starvation in the head of adult flies, but since similar means were observed between the tissues, it is most likely that the difference in outcome was due to variations in the number of samples (body 0 h n = 6 and 3 h n = 8; head 0 h n = 5 and 3 h n = 5). 

### 3.7. CG18549 Knockdown Affects the Expression of Glucagon-Like Genes

*CG18549* has been shown to be expressed in metabolic active tissues such as the fat bodies and salivary glands [35]. Together with the earlier findings about transcription binding sites linked to metabolism in the promoter region of mouse *Mfsd11* [20] as well as the altered gene expression as a response to nutrient levels, led us to the theory that *CG18549* knockdown possibly affects metabolic cues. To study this, we investigated alterations in gene expression of glucagon and insulin-like peptide and their respective receptor. The expression of *Akh*, *AkhR*, *Ilp2*, *Ilp3*, *Ilp5*, *Ilp6* and *InR* were measured in whole-body tissue samples of five- to seven-days-old Driver control (da-GAL4), RNAi control and *CG18549* knockdown flies (*CG18549* RNAi line 1), Figure 5A,B. Differences in expression were observed between the *CG18549* knockdown flies and both Driver and RNAi control flies for *Akh* (upregulated) and *AkhR* (downregulated). Expression differences against the Driver control were also measured for *Ilp5*, *Ilp6* and *InR*, Figure 5A,B. No differences were observed for *Ilp2* and *Ilp3*, Figure 5A. Interestingly, the results of the RNA sequencing, Table 2 and Appendix A, on the siblings from the same three crosses did only display similar alterations in gene expression of *Akh* and *AkhR*, suggesting that the changes in the insulin-like peptides and the insulin receptor observed from the quantitative PCR are secondary effects depending on other factors than the knockdown.

The largest effects were measured for *Akh* and *AkhR*; hence, these two genes were also run on the second *CG18549* RNAi line (*CG18549* RNAi line 2), Figure 5C, to study if the alterations were also present; hence, this strengthens the conclusion that *CG18549* affects this hormone and its receptor. Gene expression of *CG18549* in adult male flies were measured and an alteration of *Akh* were found between the RNAi control and the Driver as well as between the RNAi control and *CG18549* knockdown, while no differences were found for *AkhR*, Figure 5C. This suggests that either there are line specific effects observed or that the effects on glucagon and insulin-like linked genes are secondary and not a result of the *CG18549* knockdown. The mRNA levels are constantly fluctuating, which means that the alterations of the glucagon genes could rather be due to the time point of food intake of the flies. 

### 3.8. CG18549 Knockdown Flies Are Equally Resistant to Starvation as the Controls

The gene expression of *CG18549* has been shown to be affected by starvation ([20], while we observe no change in gene expression when limiting the study to the head. Therefore, we wanted to investigate starvation further, to study if the starvation resistance of the *CG18549* knockdown flies was affected, Figure 5 and Table 3. The driver line used for the starvation resistance assay was the da-GAL4 line. Five- to seven-days-old male flies were subjected to starvation in drosophila activity monitor systems (DAMS) where activity (beam breaks) was monitored and the last beam break of each fly was determined as the time point of death. The average last beam break of *CG18549* knockdown flies was similar to the average last beam break of the Driver control flies, but a little higher than the last beam break of the RNAi control flies. A difference between the two controls were also monitored, Figure 5D. In addition, the survival proportions were plotted to be able to study the death curves more in detail and for instance see if they are similar or for example biphasic. When comparing the survival proportions over time the RNAi control flies were observed to be more sensitive to starvation, but also the *CG18549* knockdown flies deceased faster compared with Driver control flies, Figure 5E and Table 3. In addition, the activity during starvation was plotted to study if the difference in starvation resistance could be due to differences in activity, which could indicate if the difference in starvation resistance is due to burning their nutrient storages at different paces, Figure 5F. The locomotion plot revealed that the RNAi control flies move more compared with both Driver control and *CG18549* knockdown flies, suggesting that the sensitivity to starvation could be that they use their nutrient storage faster and hence decease faster. However, in future experiments, measurements of nutrient storage and depletion are needed to validate this finding.

### 3.9. Ubiquitous Knockdown of CG18549 Has a Small Effect on Locomotion

Since *CG18549* expression is affected by nutrient status, and locomotion was found to differ between the RNAi control and the *CG18549* knockdown, it was decided to investigate the locomotion in more detail, Figure 6. Five- to seven-days-old male flies were collected for each cross: Driver control (daGAL4 driver), RNAi control and *CG18549* knockdown (*CG18549* RNAi line 1) flies and the total beam breaks were monitored and the average total beam breaks per hour was calculated. The RNAi control was once again found to be more active compared with the Driver control and the *CG18549* knockdown flies while no difference between the Driver control and the *CG18549* knockdown could be observed, Figure 6A. However, to further investigate the locomotory, beam breaks over time, specifying the activity of each genotype per hour, were analyzed. Overall, the RNAi control had a higher activity during several time-points times during the monitored time (24 h) compared with the Driver control and *CG18549* knockdown flies, supporting the difference in total activity. However, at some time points, e.g., at 13.00–13.59, 14.00–14.59 and 08.00–08.59, *CG18549* knockdown flies had altered activity compared with only the driver control flies, while at 00.00–00.59 *G18549* knockdown flies were more active than both controls, Figure 6B. 

### 3.10. Dopamine Signal Transmission Genes and the Vesicular Monoamine Transporter in CG18549 Knockdown Flies

This is the first time *CG18549* has been shown to influence locomotion; therefore, gene expression of dopamine-related genes was measured, Figure 6C–E. To do so, the relative mRNA expression of the dopamine transporter (*Dat*), dopamine receptor subunits (*Dop1R1*, *Dop1R2*, *Dop2R)*, tyrosine hydroxylase (*Ple)* and the vesicular monoamine transporter (*Vmat*) were measured via quantitative PCR and RNA sequencing in Driver controls, RNAi controls and ubiquitous *CG18549* knockdown flies. We hypothesized that there could be a difference in these genes causing the alteration in locomotion. No statistical differences were measured for *Dat*, *Dop1R1*, *Dop1R2*, *Dop2R* and *Ple*, Figure 6C. No clear differences were observed for *Vmat* either, but a trend towards that expression is reduced in the knockdown flies compared with the Driver control flies was observed; however, it was not significant, Figure 6D. Meanwhile, the RNA sequencing data showed similar findings for *Dat* and *Dop1R1* where no differences were observed between the three phenotypes. However, all the other genes investigated were found to be affected in the *CG18549* knockdown flies compared to either the Driver control flies (*Ple*), alone or compared with both controls (*Dop1R2*, *Dop2R* and *Vmat*). Interestingly, the RNA sequencing data revealed that both *Dop2R* and *Vmat* were also significantly reduced compared with both controls, while no large differences were observed between the controls, Table 2. 

To measure if the results are linked to *CG18549* knockdown, one gene that was not regulated (*Ple*) and one gene that was regulated (*Vmat*) were measured using the second RNAi line (*CG18549*
*RNAi line 2*). The gene expression of *Ple* and *Vmat* were measured in ubiquitous *CG18549* knockdown male flies and controls aged five to seven days, Figure 6E. *Ple* was found to be downregulated in both the RNAi control and the *CG18549* knockdown flies compared with the Driver control flies, while no difference was observed between the RNAi control and *CG18549* knockdown flies, a result that is consistent with the verification result of the knockdown. A small difference in *Vmat* expression was detected in *CG18549* knockdown flies compared with both controls Figure 6E. These results together with the locomotion behavior of ubiquitous *CG18549* knockdown suggest that there are differences present, and the effect on *Vmat* in the first RNAi line as well as in the second RNAi line suggest that it could be linked to the vesicular monoamine transport. However, it is also likely that there are compensatory mechanisms activated in the *CG18549* knockdown flies which cause the phenotype to not be prominent.

### 3.11. RNA Sequencing on CG18549 Knockdown Flies and Controls

RNA sequencing was performed on Driver control (da-GAL4, RNAi control and *CG18549* knockdown (*CG18549* RNAi line 1) to investigate if *CG18549* knockdown affects parts of or whole cellular pathways. No obvious pathways were pinpointed and gene expression that was affected in *CG18549* knockdown samples appeared in several different clusters, including non-coding RNA, miRNA, protein-coding genes and more, Appendix A. Instead, the RNA sequencing data, Table 2, were used to confirm the metabolism and activity related genes that were measured with qPCR in Figure 5 and Figure 6. Similar alteration patterns were observed for *CG18549* (down), *Akh* (up), *AkhR* (down), *Ilp2* (down), *Ilp3* (no change), *Dat* (no change), *Dop1R2* (no change), *Dop2R* (down) and *VMAT* (down). Meanwhile *Dop1R1* was found to be downregulated in the *CG18549* knockdown flies compared with the Driver control flies, a trend that is also observed in the qPCR data; however, it was not confirmed by statistics. *Ilp5* (qPCR: down; RNAseq: up) and *Ple* (qPCR: up-trend; RNAseq: down) were found to have the opposite regulation compared with the qPCR, Table 2. Interestingly, the RNA sequencing data on *Ple* is consistent with the qPCR data measured in the second RNAi line, Figure 6D.

### 3.12. Conditional Knockdown of CG18549 in Adults Using the Elav-GAL4:GAL80 Alters Movement and Gene Expression of the Vesicular Monoamine Transporter

If the inconsistent findings found for the ubiquitous knockdown of *CG18549* are due to compensatory mechanisms, a conditional knockdown could aid in detangling the link between *CG18549* and locomotion. Therefore, monitoring of locomotion behavior was performed using adult flies from the *CG18549* RNAi line 1 and the conditional brain cell driver elav-GAL4:GAL80. The locomotion assay was run for 48 h. During the first 24 h, no significant difference was observed, Figure 7A, but effect size estimations suggest that *CG18549* knockdown flies had altered locomotion compared with both controls. During the second day, the *CG18549* knockdown flies and RNAi controls moved more than the Driver control flies, Figure 7A. In an attempt to study if the difference in locomotion was due to differences in initiating and terminating a movement, filming, and manually scoring of starts and stops, was performed for each genotype, but no differences were observed, Figure 7B. When looking at the locomotion per hour during 24 h, it was clear that *CG18549* knockdown flies were more active compared to both controls, even if the RNAi control flies were more active than the Driver control flies, as well for 16 of the 24 h, Figure 7C. To establish the findings in the ubiquitous knockdown flies, the gene expression of *Dat*, *Dop1R1*, *Dop2R*, *Ple* and *Vmat* were measured again but in the conditional knockdown. *Vmat* was found to be downregulated in both the RNAi control and *CG18549* knockdown, while no alterations of *Dat*, *Dop1R1*, *Dop2R* and *Ple* were measured using qPCR, Figure 7C. These results suggest that the conditional knockdown produce progenies with a clearer phenotype compared with ubiquitous knockdown, making it possible that it is compensatory mechanisms that cause the less prominent phenotype in the ubiquitous knockdown. However, compensatory mechanisms would need to be investigated in a greater depth to conclude what are the mechanisms that compensate. 

### 3.13. Brain Knockdown of CG18549 Alters Locomotion in Five- to Seven-Days-Old Male Flies

*CG18549* has now been revealed to affect locomotion when using a conditional driver line specific for the brain in *D. melanogaster*. Despite conditional knockdown being a good model to study gene knockdown during latter stages of life, the GAL4:GAL80 system requires that the crosses are maintained at a higher temperature to dissolve the GAL80 from the GAL4 to activate the transcription. Higher temperature can, on its own, affect the fly; therefore, subsequently, we wanted to study the effects of embryonic brain knockdown of *CG18549* using the elav-GAL4 driver to study if the phenotype remained detectable. 

The flies from the *CG18549* knockdown and control crosses were used in a total of three locomotory runs using the DAMS, Figure 8A. *CG18549* knockdown flies were found to move more compared with the Driver control in all three runs, but *CG18549* knockdown flies were only found to move more than the RNAi control during the second run; moreover, the RNAi control moved more than the Driver control. A representative plot of total activity over time of ten flies were drawn to illustrate the difference in locomotion per hour, Figure 8B. Here it was possible to observe a difference in activity during specific timepoints, and *CG18549* knockdown flies were monitored to move more at 19.00–19.59 and 09.00–09.59 compared with both controls, while at several timepoints knockdown flies moved more than the driver control. 

### 3.14. Brain Knockdown of CG18549 and Locomotion in Elderly Flies

Age has been shown to have an effect on locomotion responses and changes in monoamine levels and metabolites in mice [38]. In fly, *Vmat* expression has been found to be of great importance to locomotive behavior [39]. Therefore, we wanted to study how age affects locomotion in *CG18549* knockdown flies that also display an already existing reduction in *Vmat* expression. An age-dependent locomotion assay was performed on flies (elav-GAL4 driver and *CG18549* RNAi line 1) aged 13 and 21 days. RNAi controls and *CG18549* knockdowns were found to be less active at an age of 13 days old compared with the Driver control, Figure 8C, while no differences in activity could be measured in 21-days-old flies, Figure 8D. When plotting activity over time for flies aged to 13 days, the Driver control was found to have twice as high activity, Figure 8E, while at an age of 21 days, all three, the Driver control, the RNAi control and *CG18549* knockdown flies moved equally, Figure 8F. However, the assay for the flies aged to 13 days contain fewer flies and the variation is larger within the groups, hence this particular result should be interpreted with caution. Taken together, the results suggest that *CG18549* knockdown and RNAi control flies have an earlier onset of slower movement compared with Driver control flies. 

### 3.15. Brain Knockdown of CG18549 Affects Stress Response in Flies

The inconsistency in movement observed, as well as the alterations in locomotion early in the experimental setups could indicate that a stress-related phenotype is present. Also, earlier findings about the locomotion phenotype, monoamine loading and *Vmat* expression formed the hypothesis that the locomotion alterations observed for *CG18549* flies are due to the low expression of *Vmat*, suggesting that the knockdown flies cannot load vesicles with enough monoamines during more active conditions such as stress. Furthermore, altered *Vmat* expression could cause other phenotypes such as anxiety [40]. Therefore, a stress assay, where the flies are exposed to bright light three times a day (time point indicated by arrows) for two days were performed on brain knockdown flies and controls, Figure 9. Flies were exposed to light at 09.00, 12.00 and 15.00 the first day and 09.00, 12.00 and 16.00 the second day. Initially, no differences were observed when plotting the total activity over time, but after the flies were stimulated with bright light, an altered number of beam breaks could be observed for the *CG18549* knockdown flies compared with both controls, Figure 9A. Interestingly, after the third stimulation with bright light, all three genotypes recovered after an hour, but three to four hours post-stimulation, the *CG18549* knockdown flies experienced a sudden drop in locomotion, which were not observed for either controls, Figure 9A, time point 18.00–18.59 until 03.00–03.59. This was observed during both days. Moreover, an increased locomotion peak was observed for the *CG18549* knockdown flies after 21.00, while the controls activity decreased as normal when following the circadian rhythm. The peak was also observed during the second day, suggesting that the *CG18549* knockdown flies are indeed not able to handle the bright light as the controls. In contrast, when analyzing the total beam breaks during both days, a trend of increased activity could be observed for both the RNAi control and the *CG18549* knockdown flies, but they were not statistically significant, Figure 9B. Moreover, to study immediate stress responses the differences in activity the hour after stimulation with bright light were analyzed. Here, an increase in activity could be observed for the *CG18549* knockdown flies, but it was only statistically significant at timepoint 13.00–13.59 of both days, Figure 9C, and the response to the bright light minimized as expected with each stimulation, which is observed on the total beam breaks at each time point. Taken together, the results suggest that *CG18549* knockdown flies change their locomotion behavior post-stimulation and had difficulties to recover several hours after the stressor. However, there is not a clear indication that *CG18549* is linked to stress-behavior and anxiety-related behavior, and further experiments are needed to establish that connection.

## 4. Discussion

MFSD11 is an evolutionary-conserved protein and related proteins are identified in both prokaryotes and eukaryotes [17,27], but still the expression and function of MFSD11 is not fully known. A couple of years ago, MFSD11 was identified as an “atypical” SLC [16], where CG18549 was suggested to be the orthologue to MFSD11. The closest relatives to MFSD11 in the human proteome are the UNC93 proteins (UNC93A and UNC93B1) [27], two proteins with very different function. UNC93A is believed to be involved in potassium fluxes [27,41,42], while UNC93B1 is established to be a transporter of toll-like receptors [43,44,45]. MFSD11 has also been showed to cluster with the SLC45 (proton:sugar cotransporter) and the SLC46 (folate transporter) superfamilies of the SLCs [17]. This suggests that MFSD11 could be a transporter of solubles, or it could have similar functions as UNC93A (regulatory subunit) and UNC93B1 (Receptor transporter and aid in membrane attachment). However, this still remains uncertain. 

Previous data suggest that the murine *Mfsd11* reacts to altered nutritional levels and longer periods of starvation [17] and its promoter contains several binding sites for transcriptional regulation that react to changes in general metabolism, but also to specific changes in amino acid and glucose metabolism [20]. *CG18549* has been found to be altered by starvation in whole-body samples from wildtype *D. melanogaster* [20]; although we did not observe a clear effect in head tissue samples from *D. melanogaster*, similar differences could be observed when comparing the whole-body and the head tissue samples. It is possible that the protective environment of the brain [46] minimizes the effect starvation has on certain genes, even if the gene has several binding sites for transcription factors, enhancers and silencer that should alter its expression as a response to nutrient levels. Moreover, it could be that the transcriptional regulation is specific to certain cell types; hence, *CG18549* was not affected in the head tissue because it does not execute the same function there. It is also possible that CG18549 does not have a specific role in starvation, but rather its expression is regulated by other genes. The results suggest that *CG18549* is a more stable gene during starvation than other atypical SLCs such as *CG4928*, the orthologue to human *UNC93A* [27].

Transcriptional changes after flies have been subjected to diets with sugar and protein imbalance have been found previous in whole-body samples from *D. melanogaster* [20]. Here, we found that *CG18549* was altered in head tissue obtained from wildtype flies fed diets with strong sugar and protein imbalances. Since *CG18549* expression was altered both during high and low sugar and/or protein, no clear link to one nutrient was observed. *CG18549* was not affected by a low calorie diet (2.5:2.5 g/dL sugar:yeast), which was observed in the body tissue [20]. The mouse *Mfsd11* has several transcription binding sites for regulation by both amino acid and glucose sensing elements [20], related bindings sites were not available in the eukaryotic promoter database [47] for *D. melanogaster*; hence, they were not searched for, but it is likely that *CG18549* possess similar sites. If *CG18549* also possess these bindings sites of different sensing elements as the human *MFSD11* and mouse *Mfsd11*, it could explain why we can measure gene expression alterations in diets that are the opposite of each other, e.g., the 2.5:40 g/dL and 40:2.5 g/dL. These results would also then suggest that *CG18549* have a more general role in the cell compared to sugar and amino acid transporters that are mainly found to be affected after a specific substrate has been altered [20,48,49]. However, the possibility that CG18549 is involved in solute transport remains, and it could be that CG18549 transport ions, e.g., hydrogen and sodium, that is needed of both amino acid and sugar transporters for proper function. Therefore, similar transcriptional regulation would be in place during different diets, since its role would be to maintain ion balance. However, it is still only speculation, but similar functions have been observed for its close relative *Unc93a* [27,32]. 

In addition, an effect on the *adipokinetic hormone* (*Akh*) and its receptor (*AkhR*) [50,51,52] in *CG18549* knockdown flies were observed, which suggest that *CG18549* could have a potential role in the metabolic pathway of glucagon. *CG18549* is highly expressed in the fat bodies [35], which could strengthen this theory. The locomotory phenotype observed of *CG18549* knockdown flies could also possibly be explained by the transcriptional changes of *Akh.* Akh is involved in locomotor activity stimulation [53] and a depletion of the Akh-producing cells have been reported to result in a decreased locomotor activity [53], suggesting that the increased *Akh* expression could promote a more active phenotype. 

*CG18549* knockdown and RNAi controls flies were found to have an earlier onset of locomotory decline that are observed with age [54]. Interestingly, the *CG18549* knockdown flies also reacted stronger to bright light stimulation compared with both controls and both locomotory activity alterations, as well as changes in the rhythm were observed, suggesting that the flies cope with that kind of stressor differently. It is therefore possible that an anxiety-related phenotype is present [55,56], but more studies are needed to confirm the speculation. *CG18549* was abundantly expressed in the larvae and adult CNS, but no double-immunostaining was performed; hence, certain neuronal and/or glial populations were not identified. This makes it difficult to give a detailed explanation to the locomotory phenotype. According to previous findings in mice [17], MFSD11 was found to overlap with anti-glutaminase [17]. Furthermore, MFSD11 was found in vesicular inhibitory amino acid transporter (VIAAT)-eGFP positive cells [17]. Taken together, the expression profile in mice suggest that MFSD11 is expressed in different neuronal subpopulations and the authors suggest that MFSD11 is expressed in both excitatory and inhibitory subpopulations [17]. 

Recently, shared behavior processes between arthropods and vertebrates were discovered, and the central complex and the basal ganglia were found to have a deep homology [57]. Dysfunctions in both these systems cause behavioral defects such as motor abnormalities, impaired memory formation and more [57]. Dysregulation in the central complex could be one explanation to why *CG18549* knockdown flies have an altered locomotion. For example, *Vmat* expression was affected by *CG18549* knockdown, and altered packaging of neurotransmitters could be a reason to the locomotion pattern. In humans, there are two VMATs (VMAT1 and VMAT2) that act as exchangers of monoamine and hydrogen, and are responsible for packaging, e.g., dopamine, serotonin, and norepinephrine in vesicles [58]. In *D. melanogaster*, there is one *Vmat* orthologue that encodes a protein responsible for packaging neurotransmitters such as dopamine, serotonin and octopamine into secretory vesicles [39,59]. Dopamine and serotonin are also vital for locomotion [60,61] and are also closely linked to stress and anxiety [55,62]. Therefore, it is possible and more likely that the locomotory phenotype that we observe were due to lowered packaging of monoamines. Furthermore, we believe that the sudden decline in activity and the long recovery time observed for the *CG18549* knockdown flies after the stressor (light) was induced, support the suggestion that vesicles are not filled with the right neurotransmitter in a normal rate; therefore, the locomotory activity curve becomes skewed. 

## 5. Conclusions

To conclude, this is the first in-depth characterization focusing on the physiological role of MFSD11 performed in the model organism *D. melanogaster*, and the study’s main findings were: (I) MFSD11 and CG18549 are predicted to have very similar secondary and tertiary structure, (II) *CG18549* was found to be abundantly expressed in both third instar larvae CNS and the adult brain, (III) *CG18549* was altered by nutrient levels, especially diets with strong sugar:protein imbalance, (IV) brain knockdown of *CG18549* affect locomotion and results in more active phenotype, (V) *CG18549* knockdown flies were found to react differently post-stimulus of the a stressor (bright light) and (VI) *Vmat* expression was reduced in the *CG18549* knockdown flies. These results conclude that *CG18549* is evolutionary conserved, abundantly expressed in more species than mammals, the gene expression is affected by what the flies eat and that there is a link between *CG18549* expression, locomotion and *Vmat*. However, to establish this link, e.g., the levels of neurotransmitters need to be measured, which is something that has not been undertaken here. The exact mechanisms behind all observations are not fully understood, but our study provides new insights into the expression and function of *CG18549*. Clearly, these results, among others about this orphan transporter, provide a strong example as to why it is vital to fully characterize them, and through that, gain knowledge about the body during normal conditions and disease.

## Figures and Tables

**Figure 1 insects-12-01024-f001:**
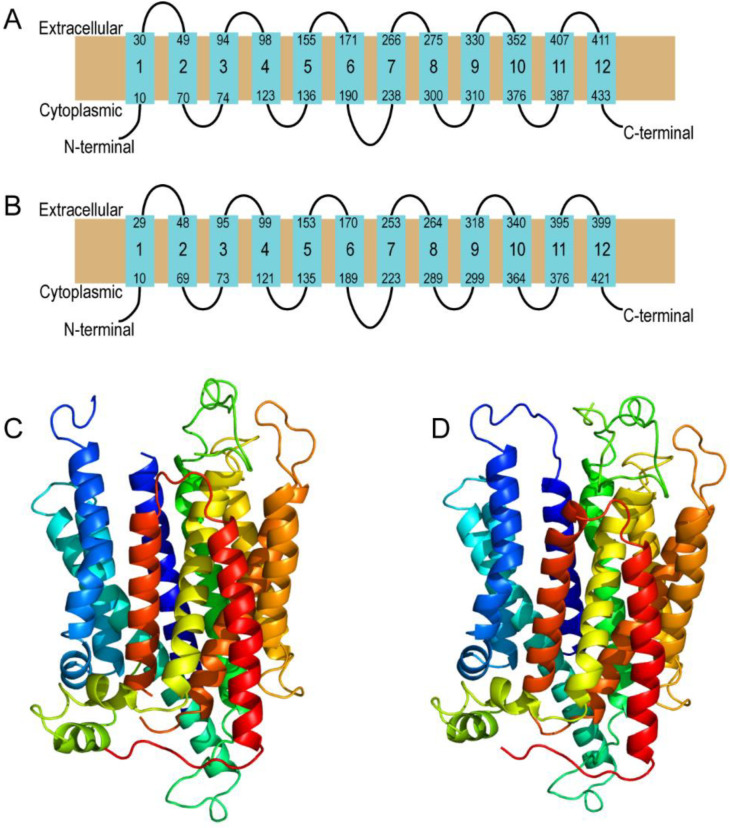
Prediction of secondary and tertiary structure of human MFSD11 and *D. melanogaster* CG18549. Homology modeling and predictions were performed using Phyre2 [23]. Both proteins were modeled against unc-93 homolog b1 from *Mus musculus* (PDB id c7c77B) [24]. (**A**) Secondary structure of human MFSD11. It was modeled with 11% identity, 100% confidence and 94% amino acid coverage. (**B**) Secondary structure of *D. melanogaster* CG18549. It was modeled with 12% identity, 100% confidence and 97% amino acid coverage. (**C**) Tertiary structure of human MFSD11 and (**D**) *D. melanogaster* CG18549. The gradient of the tertiary structures displays helices from N-terminal (dark blue) to C-terminal (red). Unconnected loops in the predictions are parts of the 3D structure that are modeled with uncertainties and, therefore, they cannot be fully modeled.

**Figure 2 insects-12-01024-f002:**
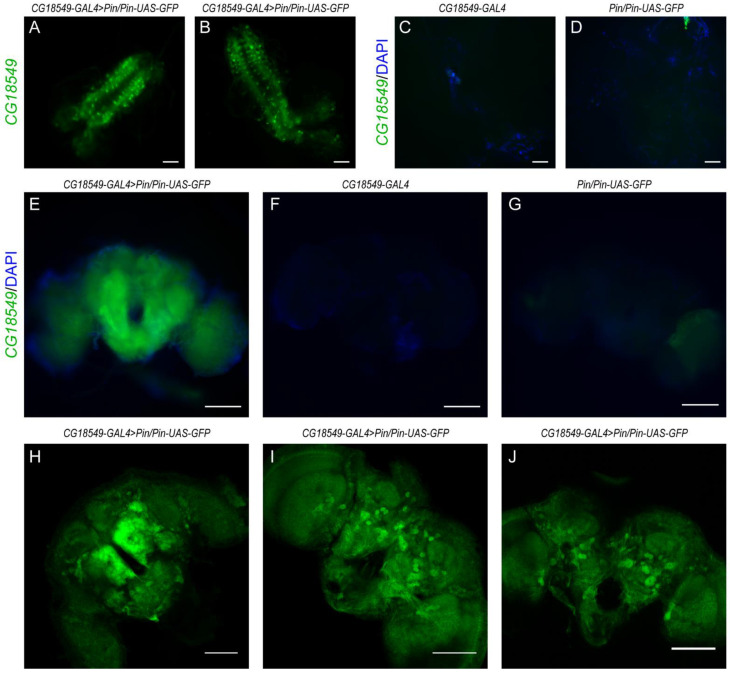
*CG18549* gene expression images of the third instar larvae CNS and the adult brain. (**A**–**D**) Dissected brains and ventral nerve cord from third instar larvae, *CG18549* expression was visualized with green fluorescence and blue (DAPI) marks the nucleus, scale bar 20 µm. (**A**,**B**) *CG18549-GAL4* > *Pin/Pin-UAS-GFP*, (**C**) CG18549-GAL4 and (**D**) Pin/Pin-UAS-GFP larvae. (**E**–**G**) Dissected brains from seven-days-old adult flies, *CG18549* expression was visualized with green fluorescence and blue (DAPI) marks the nucleus, scale bar 100 µm. (**E**) *CG18549-GAL4* > *Pin/Pin-UAS-GFP*, (**F**) CG18549-GAL4 and (**G**) Pin/Pin-UAS-GFP adult flies. (**H**–**J**) 2-photon, high resolution images of *CG18549-GAL4* > *Pin/Pin-UAS-GFP* (n = 3) seven-days-old flies. *CG18549* expression was visualized with green fluorescence and blue (DAPI) marks the nucleus, scale bar 100 µm. Brains were imaged with Z-stacks and the pictures were merged displaying the average intensity. (**H**) Picture include 60 (1 µm) slides, (**I**) picture include 68 (1 µm) slides, (**J**) Picture include 29 (1 µm) slides.

**Figure 3 insects-12-01024-f003:**
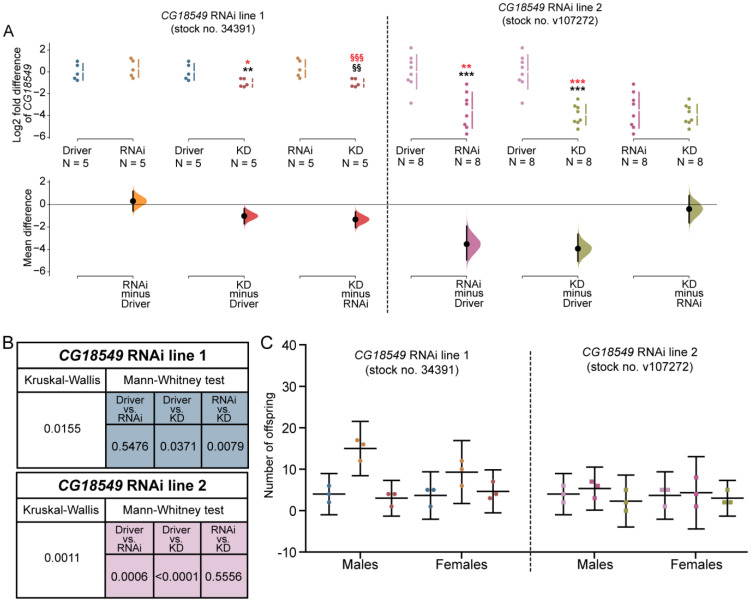
Validation of RNAi line functionality and counting of F1 progenies. *CG18549* knockdown was verified for two *CG18549* RNAi lines: y[1] sc[*] v[1]; P{y[+t7.7] v[+t1.8] = TRiP.HMS01385}attP2 (*CG18549* RNAi line 1, Stock no. 39341) and P{KK102196}VIE-260B (*CG18549* RNAi line 2, Stock no. v107272) using the ubiquitous driver da-GAL4. Gene expression was measured with qPCR, log2 fold change was calculated according to the delta Ct method using three reference genes (*Actin42a*, *Rpl11*, *Rp49*), the Driver control was set as control. Kruskal–Wallis with Dunn’s comparison against the control was used to initially analyze differences, exact p-values were calculated using multiple corrected Mann–Whitney as a post hoc test * or § *p* < 0.0491, ** or §§ *p* < 0.0099, *** or §§§ *p* < 0.0001 (* = against Driver control, § = against RNAi control). (**A**) The Log2 fold mean differences for 6 comparisons are shown in the above Cumming estimation plot. The raw data is plotted on the upper axes; each mean difference is plotted on the lower axes as a bootstrap sampling distribution. Mean differences are depicted as dots; 95% confidence intervals are indicated by the ends of the vertical error bars. The effect sizes and CIs are reported above as: effect size [CI width lower bound; upper bound]. *CG18549* RNAi line 1: Kruskal–Wallis *p* = 0.0155, Mann–Whitney (MW) *CG18549* knockdown * *p* = 0.0317, §§ *p* = 0.0079, the unpaired mean difference between Driver ctrl and RNAi ctrl is 0.305 [95.0% CI −0.57, 1.19], *p*-value of the two-sided permutation *t*-test is 0.524; Driver ctrl and *CG18549* knockdown is −1.02 [95.0% CI −1.71, −0.382], *p*-value of the two-sided permutation *t*-test is 0.021; RNAi ctrl and *CG18549* knockdown is −1.33 [95.0% CI −2.05, −0.659], *p*-value of the two-sided permutation *t*-test is 0.0004. *CG18549* RNAi line 2: Kruskal–Wallis < 0.0001, MW *CG18549* knockdown *p* ≤ 0.0001, RNAi ctrl *p* = 0.0006, the unpaired mean difference between Driver ctrl and RNAi ctrl is −3.53 [95.0% CI −4.93, −1.94], *p*-value of the two-sided permutation *t*-test is 0.0012; Driver ctrl and *CG18549* knockdown is −3.93 [95.0% CI −5.05, −2.65], *p*-value of the two-sided permutation *t*-test is 0.0006; RNAi ctrl and *CG18549* knockdown is −0.404 [95.0% CI −1.62, 0.789], *p*-value of the two-sided permutation *t*-test is 0.556. (in red effect size differences * = significant difference against the Driver control, § = significant difference against the RNAi control). (**B**) Summary of *p*-values in the statistical analysis. (**C**) F1 progenies were counted from both RNAi lines and controls, the males and females of F1 progenies are shown in the above scatter plot.

**Figure 4 insects-12-01024-f004:**
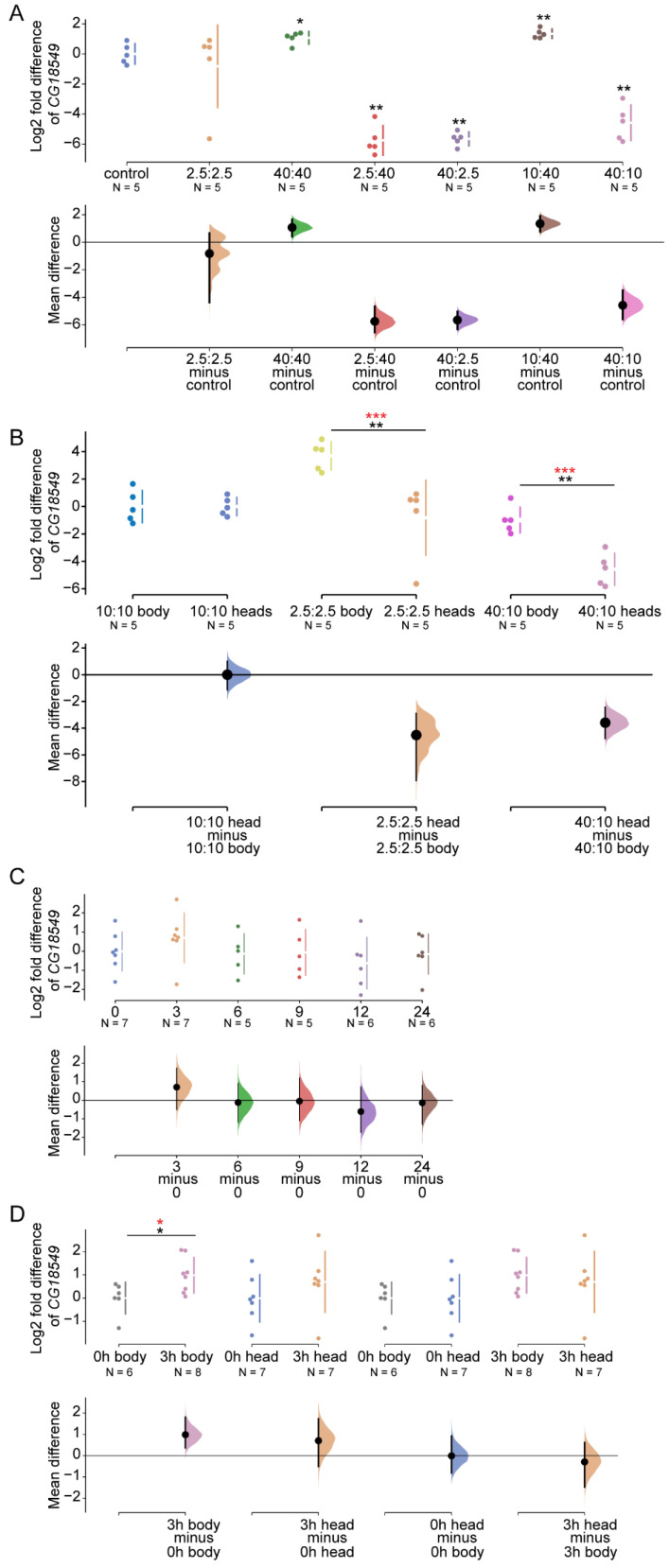
*CG18549* expression alterations as a response to sugar and protein imbalanced diets, and starvation. qPCR was used to measure the gene expression of *CG18549* in heads from five- to seven-days-old CSORC flies (n = 5) fed diets with different sugar and yeast ratio: 10:10 g/dL (control), 2.5:2.5 g/dL, 40:40 g/dL, 2.5:40 g/dL, 10:40 g/dL, 40:2.5 g/dL and 40:10 g/dL diets or subjected to 0 (n = 7), 3 (n = 7), 6 (n = 5), 9 (n = 5), 12 (n = 6) and 24 (n = 6) hours of starvation. The log2 fold change was calculated with the delta Ct method against three reference genes (*Actin42a*, *Rpl11*, *Rp49*), the Driver control was set as control. Kruskal–Wallis with Dunn’s comparison against the control was used to initially analyze differences, exact *p*-values were calculated using multiple corrected Mann–Whitney as a post hoc test * *p* < 0.0489, ** *p* < 0.0099, *** *p* < 0.0001. (**A**) Log2 fold mean differences of *CG18549* for 6 comparisons against the shared control 10:10 g/dL sugar:yeast are shown in the a Cumming estimation plot. The raw data is plotted on the upper axes. On the lower axes, mean differences are plotted as bootstrap sampling distributions. Each mean difference is depicted as a dot. Each 95% confidence interval is indicated by the ends of the vertical error bars. The effect sizes and CIs are reported above as: effect size [CI width lower bound; upper bound]. Kruskal–Wallis *p* = 0.0001, the unpaired mean difference between control and 2.5:2.5 g/dL is Mann–Whitney (MW) *p* = 0.6905, effect size −0.822 [95.0% CI −4.38, 0.664], *p*-value of the two-sided permutation t-test is 0.822; control and 40:40 g/dL is MW *p* = 0.0317, effect size 1.07 [95.0% CI 0.408, 1.65], *p*-value of the two-sided permutation *t*-test is 0.019; control and 2.5:40 g/dL is MW *p* = 0.0079, effect size −5.75 [95.0% CI −6.55, −4.66], *p*-value of the two-sided permutation *t*-test is 0.0; control and 40:2.5 g/dL is MW *p* = 0.0079, effect size −5.65 [95.0% CI −6.32, −5.04], *p*-value of the two-sided permutation *t*-test is 0.003; control and 10:40 g/dL is MW *p* = 0.0079, effect size 1.35 [95.0% CI 0.758, 1.91], *p*-value of the two-sided permutation *t*-test is 0.0022; control and 40:10 g/dL is MW *p* = 0.0079, effect size −4.58 [95.0% CI −5.59, −3.49], *p*-value of the two-sided permutation *t*-test is 0.0018. Previous published data [20] was, with permission of all authors, reanalyzed and plotted against the results obtained from head tissue. (**B**) *CG18549* expression in body and head: 10:10 g/dL, 2.5:2.5 g/dL and 40:10 g/dL. The mean differences for 3 comparisons are shown in the above Cumming estimation plot with same settings as in (**B**). Kruskal–Wallis *p* = 0.0004, the unpaired mean difference between 10:10 g/dL in body and 10:10 g/dL in head is MW *p* = 0.8413, effect size 1.4 × 10^−6^ [95.0% CI −1.13, 1.0], *p*-value of the two-sided permutation *t*-test is 0.997; 2.5:2.5 g/dL in body and 2.5:2.5 g/dL in head is MW *p* = 0.0079, effect size −4.52 [95.0% CI −7.92, −2.9], *p*-value of the two-sided permutation *t*-test is 0.0; 40:10 g/dL in body and 40:10 g/dL in head is MW *p* = 0.0079, effect size −3.59 [95.0% CI −4.78, −2.44], *p*-value of the two-sided permutation *t*-test is 0.0. (**C**) The mean Log2 fold differences of *CG18549* for 5 comparisons against the shared control 0 are shown in the above Cumming estimation plot. The raw data is plotted on the upper axes. On the lower axes, mean differences are plotted as bootstrap sampling distributions. Each mean difference is depicted as a dot. Each 95% confidence interval is indicated by the ends of the vertical error bars. The effect sizes and CIs are reported above as: effect size [CI width lower bound; upper bound]. Kruskal Wallis *p* = 0.5194, the unpaired mean difference between 0 and 3 is 0.705 [95.0% CI −0.505, 1.74], *p*-value of the two-sided permutation t-test is 0.269; 0 and 6 is −0.124 [95.0% CI −1.19, 0.912], *p*-value of the two-sided permutation *t*-test is 0.844; 0 and 9 is −0.0533 [95.0% CI −1.1, 1.19], *p*-value of the two-sided permutation *t*-test is 0.928; 0 and 12 is −0.611 [95.0% CI −1.73, 0.716], *p*-value of the two-sided permutation *t*-test is 0.366; 0 and 24 is −0.139 [95.0% CI −1.29, 0.793], *p*-value of the two-sided permutation *t*-test is 0.804. Previous published data [20] was, with permission of all authors, reanalyzed and plotted against the results obtained from head tissue. (**D**) *CG18549* expression in body and head: 0 h and 3 h of starvation, the mean difference for 4 comparisons of *CG18549* expression are shown in the above Cumming estimation plot. The effect sizes and CIs are reported above as: effect size [CI width lower bound; upper bound]. Kruskal–Wallis *p* = 0.0325, the unpaired mean difference between 0 h body and 3 h body is MW *p* = 0.0293, effect size 0.987 [95.0% CI 0.371, 1.81], *p*-value of the two-sided permutation *t*-test is 0.0248; 0 h head and 3 h head is MW *p* = 0.2593, effect size 0.705 [95.0% CI −0.505, 1.74], *p*-value of the two-sided permutation *t*-test is 0.268; 0 h body and 0 h head is MW *p* = 0.7308, effect size −0.0074 [95.0% CI −0.806, 0.931], *p*-value of the two-sided permutation *t*-test is 0.988; 3 h body and 3 h head is MW *p* = 0.8665, effect size −0.29 [95.0% CI −1.48, 0.622], *p*-value of the two-sided permutation *t*-test is 0.622.

**Figure 5 insects-12-01024-f005:**
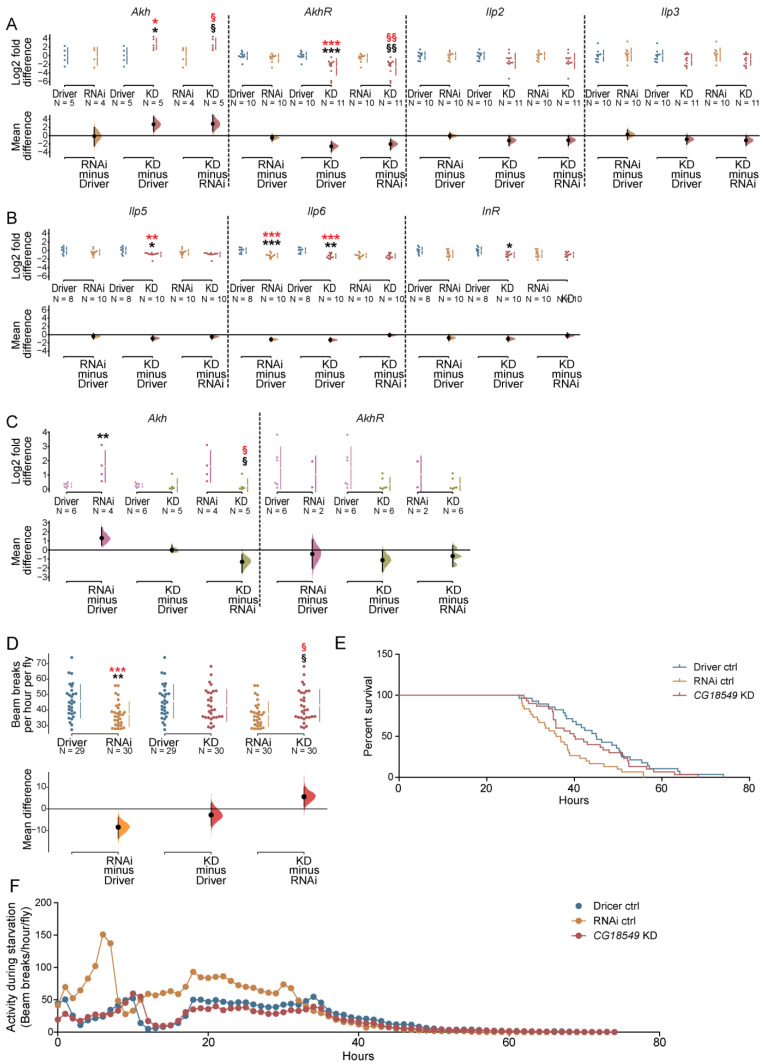
mRNA expression of glucagon and insulin like peptides and receptors, and starvation resistance in *CG18549* knockdown flies. The ubiquitous da-GAL4 driver, w^1118^ and *CG18549* RNAi line 1 and 2 were used. Gene expression of *Akh*, *AkhR*, *Ilp2*, *Ilp3*, *Ilp5*, *Ilp6* and *InR* were measured with qPCR, log2 fold changes were calculated according to the delta Ct method using three reference genes (*Actin42a*, *Rpl11*, *Rp49*) for *CG18549* RNAi line 1 and controls, while one reference gene (*Actin42a*) was used for *CG18549* RNAi line 2 and controls. The Driver control was set as control. Kruskal–Wallis with Dunn’s comparison against the control was used to initially analyze differences, exact *p*-values were calculated using multiple corrected Mann–Whitney as a post hoc test * or § *p* < 0.0491, ** or §§ *p* < 0.0099, *** or §§§ *p* < 0.0001 (in red effect size differences * = significant difference against the Driver control, § = significant difference against the RNAi control). (**A**,**B**) Data of *CG18549* RNAi line 1 (* = significant difference against the Driver control, § = significant difference against the RNAi control. The mean differences for 21 comparisons are shown in the above Cumming estimation plot. The raw data is plotted on the upper axes; each mean difference is plotted on the lower axes as a bootstrap sampling distribution. Mean differences are depicted as dots; 95% confidence intervals are indicated by the ends of the vertical error bars: *Akh* (RNAi ctrl minus Driver ctrl −0.127 [95.0% CI −2.48, 1.95] *p* = 0.886; *CG18549* knockdown minus Driver ctrl 2.75 [95.0% CI 1.03, 4.6] *p* = 0.0254; *CG18549* knockdown minus RNAi ctrl 2.88 [95.0% CI 1.07, 5.16] *p* = 0.0268); *AkhR* (RNAi ctrl minus Driver ctrl −0.55 [95.0% CI −1.34 *p* = 0.197; *CG18549* knockdown minus Driver ctrl −2.6 [95.0% CI −3.96, −1.56] *p* = 0.0006; *CG18549* knockdown minus RNAi ctrl is −2.05 [95.0% CI −3.42, −0.971] *p* = 0.0036); *Ilp2* (RNAi ctrl minus Driver ctrl −0.058 [95.0% CI −0.799, 0.665] *p* = 0.888; *CG18549* knockdown minus Driver ctrl −1.19 [95.0% CI −2.46, −0.223] *p* = 0.0542; *CG18549* knockdown minus RNAi ctrl −1.13 [95.0% CI −2.4, −0.166] *p* = 0.0656); *Ilp3* (RNAi ctrl minus Driver ctrl 0.263 [95.0% CI −0.946, 1.41] *p* = 0.675; *CG18549* knockdown minus Driver ctrl −0.904 [95.0% CI −2.05, 0.0776] *p* = 0.126; *CG18549* knockdown minus RNAi ctrl −1.17 [95.0% CI −2.33, −0.051]. *p* = 0.0712); *Ilp5* (RNAi ctrl minus Driver ctrl −0.406 [95.0% CI −1.05, 0.281] *p* = 0.266; *CG18549* knockdown minus Driver ctrl −0.907 [95.0% CI −1.55, −0.339] *p* = 0.0098; *CG18549* knockdown minus RNAi ctrl −0.501 [95.0% CI −1.12, −0.068] *p* = 0.0946); *Ilp6* (RNAi ctrl minus Driver ctrl is −1.16 [95.0% CI −1.64, −0.667] *p* = 0.001; *CG18549* knockdown minus Driver ctrl −1.28 [95.0% CI −1.75, −0.749] *p* = 0.0004; *CG18549* knockdown minus RNAi ctrl −0.117 [95.0% CI −0.519, 0.318] *p* = 0.615); *InR* (RNAi ctrl minus Driver ctrl 0.777 [95.0% CI −1.47, −0.0428] *p* = 0.0628; *CG18549* knockdown minus Driver ctrl −1.0 [95.0% CI −1.61, −0.375] *p* = 0.0084; *CG18549* knockdown minus RNAi ctrl −0.223 [95.0% CI −0.8, 0.415] *p* = 0.503); *Akh* (up, Kruskal–Wallis *p* = 0.0340, Mann–Whitney (MW) * *p* = 0.0317, § *p* = 0.0317), *AkhR* (down, Kruskal–Wallis *p* = 0.0005, MW * *p* = 0.0006, § *p* = 0.0035), *Ilp2* (Kruskal–Wallis *p* = 0.1398), *Ilp3* (Kruskal–Wallis *p* = 0.2033), *Ilp5* (*CG18549* knockdown, down, Kruskal–Wallis *p* = 0.0566, MW * *p* = 0.0367), *Ilp6* (*CG18549* knockdown, down, Kruskal–Wallis *p* = 0.0015, MW * *p* = 0.0019; RNAi control, down * *p* = 0.0009) and *InR* (*CG18549* knockdown, down, Kruskal–Wallis *p* = 0.0395, MW * *p* = 0.0117). (**C**) Data of *CG18549* RNAi line 2; Log2 fold change of *CG18549* in Driver control, RNAi control and *CG18549* knockdown flies (* = significant difference against the Driver control, § = significant difference against the RNAi control, in red effect size differences * = significant difference against the Driver control, §= significant difference against the RNAi control), Cumming estimation plot illustrating the effects size with the same setup as in A and B. *Akh*: Kruskal–Wallis *p* = 0.0064, effect size RNAi ctrl minus Driver ctrl 1.31 [95.0% CI 0.552, 2.47] *p* = 0.0914; CG18549 knockdown minus Driver ctrl −0.00567 [95.0% CI −0.264, 0.535] *p* = 0.984; CG18549 knockdown minus RNAi ctrl −1.31 [95.0% CI −2.49, −0.497] *p* = 0.0264; *AkhR*: Kruskal–Wallis *p* = 0.1819, effect size RNAi ctrl minus Driver ctrl −0.45 [95.0% CI −2.02, 1.13] *p* = 0.6, CG18549 knockdown minus Driver ctrl −1.13 [95.0% CI −2.4, −0.118] *p* = 0.103, CG18549 knockdown minus RNAi ctrl −0.678 [95.0% CI −1.84, 0.47], *p* = 0.0734): *Akh* (* *p* = 0.0095, § *p* = 0.05), *AhkR* (no change). (**D**–**F**) Five- to seven-days-old male flies from Driver control, RNAi control and CG18549 knockdown (*da-GAL4* driver, n = 30) were transferred to glass tubes, prepared with 1% agarose, and contained in the DAMS until the last beam break. Last beam crossing per fly was defined as the timepoint of death. Differences were calculated using Kruskal–Wallis with Dunn’s comparison and/or Mann–Whitney as a post hoc test to calculate exact *p*-values * *p* < 0.0492, ** *p* < 0.0099, *** *p* < 0.0001 (* = significant difference against the Driver control, § = significant difference against the RNAi control, in red effect size differences * = significant difference against the Driver control, § = significant difference against the RNAi control). (**D**) The mean difference of starvation resistance is presented in the above Cumming estimation plot. The raw data is plotted on the upper axes; each mean difference is plotted on the lower axes as a bootstrap sampling distribution. Mean differences are depicted as dots; 95% confidence intervals are indicated by the ends of the vertical error bars. The unpaired mean difference between Driver ctrl and RNAi ctrl is −8.41 [95.0% CI −13.4, −3.82], *p*-value of the two-sided permutation *t*-test is 0.0006; Driver ctrl and *CG18549* knockdown is −2.78 [95.0% CI −7.8, 2.64], *p*-value of the two-sided permutation *t*-test is 0.321; RNAi ctrl and *CG18549* knockdown is 5.63 [95.0% CI 1.01, 10.1], *p*-value of the two-sided permutation t-test is 0.0206. (**E**) A survival proportion plot, Log-rank (Mantel-Cox) test (conservative) *p* = 0.0040, and (**F**)) a point-connection line graph to illustrate the survival over time and locomotion.

**Figure 6 insects-12-01024-f006:**
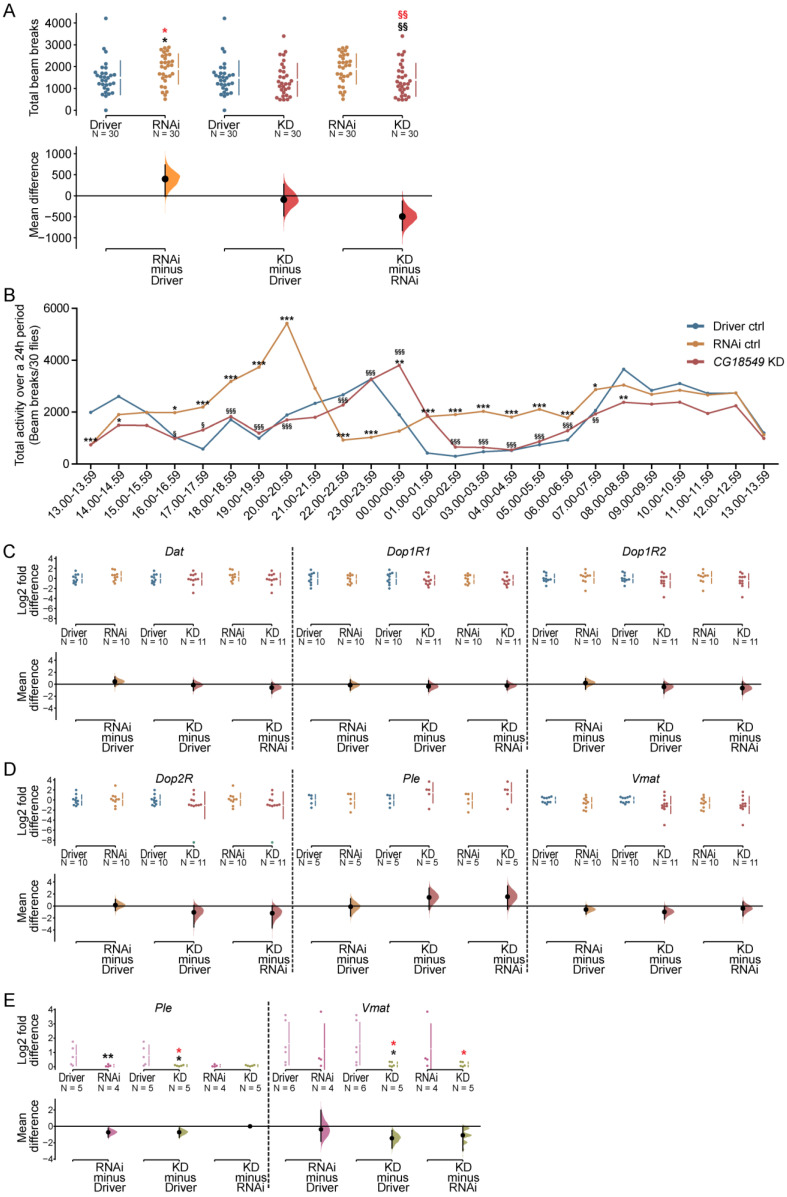
Ubiquitous knockdown of *CG18549* and locomotion monitoring and transcriptional measurements of Dopamine signal transmission genes and the vesicular monoamine transporter in *CG18549* knockdown flies. The *CG18549* RNAi line 1 was used in this setup. F1 progenies from Driver control, RNAi control and CG18549 knockdown (n = 30) were transferred to glass tubes with enriched Jazz mix standard food and contained in the DAMS for 24 h to record the activity level (locomotion, beam breaks). Differences were calculated using Kruskal–Wallis with Dunn’s comparison and/or Mann–Whitney (MW) as a post hoc test to calculate exact *p*-values * or § *p* < 0.0492, ** or §§ *p* < 0.0099, *** or §§§ *p* < 0.0001 (* = significant difference against the Driver control, § = significant difference against the RNAi control, in red effect size differences *= significant difference against the Driver control, § = significant difference against the RNAi control). (**A**) Mean differences for 3 comparisons are shown in the above Cumming estimation plot. The raw data is plotted on the upper axes; each mean difference is plotted on the lower axes as a bootstrap sampling distribution. Mean differences are depicted as dots; 95% confidence intervals are indicated by the ends of the vertical error bars. Kruskal–Wallis *p* = 0.0112; RNAi ctrl mins Driver ctrl: MW *p* = 0.0225, effect size 4 × 10^2^ [95.0% CI −12.8, 7.37 × 10^2^] *p* = 0.04; CG18549 knockdown minus Driver ctrl: MW *p* = 0.3061, effect size −91.4 [95.0% CI −4.76 × 10^2^, 2.78 × 10^2^] *p* = 0.654; CG18549 knockdown minus RNAi ctrl: MW *p* = 0.0075, effect size is −4.92 × 10^2^ [95.0% CI −8.27 × 10^2^, −1.24 × 10^2^] *p* = 0.0124). (**B**) Total activity over time. (**C**–**E**) The ubiquitous da-GAL4 driver, w^1118^ and *CG18549* RNAi line 1 and 2 were used. Gene expressions of *Dat*, *Dop1R1*, *Dop1R2*, *Dop2R*, *Ple* and *Vmat* were measured using qPCR, log2 fold changes were calculated according to the delta Ct method using three reference genes (*Actin42a*, *Rpl11*, *Rp49*) for *CG18549* RNAi line 1 and controls, while one reference gene (*Actin42a*) was used for *CG18549* RNAi line 2 and controls. The Driver control was set as control. Kruskal–Wallis with Dunn’s comparison against the control was used to initially analyze differences, exact p-values were calculated using multiple corrected Mann–Whitney as a post hoc test * *p* < 0.0491, ** *p* < 0.0099, *** *p* < 0.0001 (* = significant difference against the Driver control, § = significant difference against the RNAi control, in red effect size differences * = significant difference against the Driver control, § = significant difference against the RNAi control). (**C**,**D**) Data of *CG18549* RNAi line 1. The mean differences for 18 comparisons are shown in the above Cumming estimation plot. The raw data is plotted on the upper axes; each mean difference is plotted on the lower axes as a bootstrap sampling distribution. Mean differences are depicted as dots; 95% confidence intervals are indicated by the ends of the vertical error bars. (**C**) *Dat:* Kruskal–Wallis *p* = 0.5425, effect size unpaired mean difference RNAi ctrl minus Driver ctrl: 0.439 [95.0% CI −0.333, 1.21] *p* = 0.295; *CG18549* knockdown minus Driver ctrl: −0.128 [95.0% CI −1.09, 0.64] *p* = 0.786; *CG18549* knockdown minus RNAi ctrl: −0.567 [95.0% CI −1.52, 0.239] *p* = 0.247); *Dop1R1:* Kruskal–Wallis *p* = 0.7009, effect size unpaired mean difference RNAi ctrl minus Driver ctrl: −0.142 [95.0% CI −0.984, 0.707] *p* = 0.76; *CG18549* knockdown minus Driver ctrl −0.363 [95.0% CI −1.2, 0.582] *p* = 0.43; *CG18549* knockdown minus RNAi ctrl: is −0.221 [95.0% CI−0.922, 0.574] *p* = 0.582); *Dop1R2:* Kruskal–Wallis *p* = 0.3470, effect size unpaired mean difference RNAi ctrl minus Driver ctrl: 0.192 [95.0% CI −0.822, 0.929] *p* = 0.672; *CG18549* knockdown minus Driver ctrl −0.455 [95.0% CI −1.52, 0.369] *p* = 0.392; *CG18549* knockdown minus RNAi ctrl: −0.647 [95.0% CI −1.7, 0.413] *p* = 0.269). (**D**) *Dop2R:* Kruskal–Wallis *p* = 0.2889, effect size unpaired mean difference RNAi ctrl minus Driver ctrl: 0.141 [95.0% CI −0.768, 1.09] *p* = 0.782; *CG18549* knockdown minus Driver ctrl −1.06 [95.0% CI −3.5, 0.0975] *p* = 0.278; *CG18549* knockdown minus RNAi ctrl: −1.2 [95.0% CI −3.65, 0.0492] *p* = 0.222); *Ple:* Kruskal–Wallis *p* = 0.2276, effect size unpaired mean difference RNAi ctrl minus Driver ctrl: −0.115 [95.0% CI −1.7, 1.23] *p* = 0.927; *CG18549* knockdown minus Driver ctrl 1.43 [95.0% CI −0.615, 2.98] *p* = 0.188; *CG18549* knockdown minus RNAi ctrl: 1.55 [95.0% CI −0.584, 3.33] *p* = 0.22); *Vmat:* Kruskal–Wallis *p* = 0.1780, effect size unpaired mean difference RNAi ctrl minus Driver ctrl: −0.585 [95.0% CI −1.35, 0.106] *p* = 0.145; *CG18549* knockdown minus Driver ctrl −0.991 [95.0% CI −2.19, −0.115] *p* = 0.0804; *CG18549* knockdown minus RNAi ctrl: −0.406 [95.0% CI −1.67, 0.658] *p* = 0.549). (E) Data of *CG18549* RNAi line 2. Cummings estimation plot with same setup as in B and C: *Ple* (Kruskal–Wallis *p* = 0.0212; MW *p* = 0.0061, effect size RNAi ctrl minus Driver ctrl: MW −0.725 [95.0% CI −1.33, −0.215] *p* = 0.0568; *CG18549* knockdown minus Driver ctrl: MW *p* = 0.0101, effect size −0.716 [95.0% CI −1.35, −0.2] *p* = 0.0252; *CG18549* knockdown minus RNAi ctrl: MW *p* = 1, effect size 0.00867 [95.0% CI −0.0823, 0.0645] *p* = 0.883); *Vmat* (RNAi ctrl minus Driver ctrl: MW *p* = 0.0823, effect size −0.362 [95.0% CI −1.84, 1.98] *p* = 0.687; *CG18549* knockdown minus Driver ctrl: MW *p* = 0.0411, effect size −1.45 [95.0% CI −2.68, −0.497] *p* = 0.0394; *CG18549* knockdown minus RNAi ctrl: MW *p* = 0.0823, effect size−1.09 [95.0% CI −2.99, −0.11] *p* = 0.035.

**Figure 7 insects-12-01024-f007:**
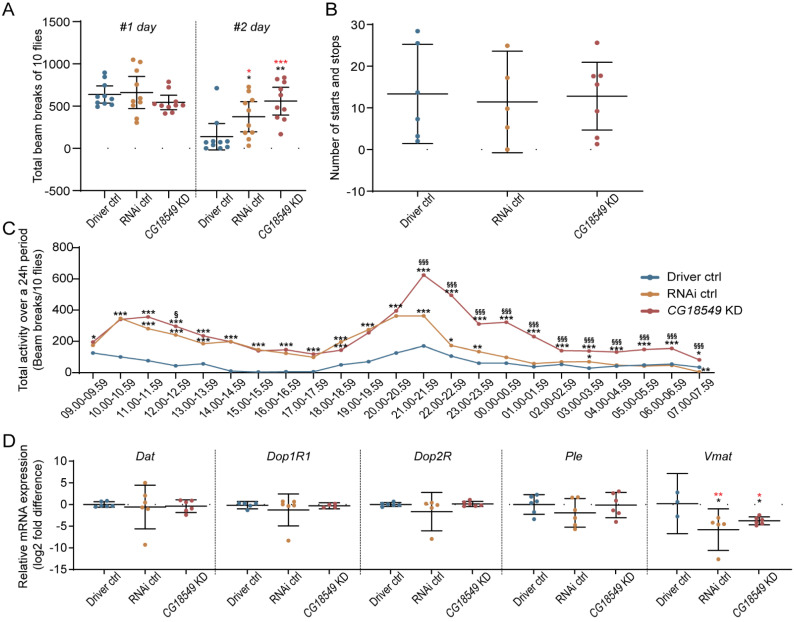
Locomotion of *CG18549* RNAi line 1 using the elav-GAL4:GAL80 driver. Conditional (onset one day post-eclosion) brain *CG18549* knockdown was performed, and activity was measured in the DAMS system. Five- to seven-days-old male flies (n = 10) were transferred to glass tubes prepared with standard Jazz mix food for 48 h. Differences were calculated using Kruskal–Wallis with Dunn’s comparison and/or Mann–Whitney as a post hoc test to calculate exact *p*-values * or § *p* < 0.0492, ** or §§ *p* < 0.0099, *** or §§§ *p* < 0.0001 (* = significant difference against the Driver control, § = significant difference against the RNAi control, in red effect size differences * = significant difference against the Driver control, § = significant difference against the RNAi control). (**A**) Scatter plot displaying the total beam breaks of run 1 (Kruskal–Wallis *p* = 0.1792) and run 2 (Kruskal–Wallis *p* = 0.0023, Mann–Whitney *CG18549* knockdown * *p* = 0.0017, RNAi ctrl * *p* = 0.0211, Mean difference for 3 comparisons were calculated with 95% confidence intervals. (in red effect size differences * = significant difference against the Driver control, § = significant difference against the RNAi control): day #1 unpaired mean difference RNAi ctrl minus Driver ctrl: 22.3 [95.0% CI −1.54 × 10^2^, 1.97 × 10^2^] *p* = 0.811; *CG18549* knockdown minus Driver ctrl: −94.7 [95.0% CI −2.02 × 10^2^, 14.8] *p* = 0.127; *CG18549* knockdown minus RNAi ctrl: −1.17 × 10^2^ [95.0% CI −2.97 × 10^2^, 52.8] *p* = 0.225), day #2 unpaired mean difference RNAi ctrl minus Driver ctrl: 2.36 × 10^2^ [95.0% CI 4.7, 3.98 × 10^2^] *p* = 0.039; *CG18549* knockdown minus Driver ctrl: 4.21 × 10^2^ [95.0% CI 1.89 × 10^2^, 5.71 × 10^2^] *p* = 0.001; *CG18549* knockdown minus RNAi ctrl: 1.85 × 10^2^ [95.0% CI −18.6, 3.79 × 10^2^] *p* = 0.0908. (**B**) Number of starts and stops for each genotype. (**C**) Total activity plotted over time (* = statistical difference against Driver control, § = statistical difference against RNAi control, * *p* < 0.05, ** *p* < 0.001, *** *p* < 0.0001). (**D**) Gene expressions of *Dat*, *Dop1R1*, *Dop2R*, *Ple* and *Vmat* were measured using qPCR, log2 fold changes were calculated according to the delta Ct method using one reference genes (*Actin42a*), the Driver control was set as control. Kruskal–Wallis with Dunn’s comparison against the control was used to initially analyze differences, exact *p*-values were calculated using multiple corrected Mann–Whitney as a post hoc test * *p* < 0.0491, ** *p* < 0.0099, *** *p* < 0.0001 (* = significant difference against the Driver control, §= significant difference against the RNAi control. No differences were measured for *Dat*, *Dop1R1*, *Dop2R and Ple*, while *Vmat* was found to be downregulated in the RNAi ctrl (Mann–Whitney * *p* = 0.0357) and *CG18549* knockdown (Mann–Whitney * *p* = 0.0476) flies. *Dat* (Kruskal–Wallis *p* = 0.9107, unpaired mean difference RNAi ctrl minus Driver ctrl: −0.591 [95.0% CI −5.12, 2.22] *p* = 0.84; *CG18549* knockdown minus Driver ctrl: is −0.387 [95.0% CI −1.57, 0.631] *p* = 0.547; *CG18549* knockdown minus RNAi ctrl: 0.205 [95.0% CI −2.63, 4.82] *p* = 0.938), *Dop1R1* (Kruskal–Wallis *p* = 0.6546, unpaired mean difference RNAi ctrl minus Driver ctrl: is −1.1 [95.0% CI −5.33, 0.647] *p* = 0.86; *CG18549* knockdown minus Driver ctrl: is −0.148 [95.0% CI −0.683, 0.644] *p* = 0.741; *CG18549* knockdown minus RNAi ctrl: 0.948 [95.0% CI −0.771, 4.06] *p* = 0.945), *Dop2R* (Kruskal–Wallis *p* = 0.7604, unpaired mean difference RNAi ctrl minus Driver ctrl: −1.66 [95.0% CI −6.45, 0.109] *p* = 0.234; *CG18549* knockdown minus Driver ctrl: 0.125 [95.0% CI −0.346, 0.686] *p* = 0.676; *CG18549* knockdown minus RNAi ctrl: 1.78 [95.0% CI −0.0278, 6.58] *p* = 0.178), *Ple* (Kruskal–Wallis *p* = 0.4785, unpaired mean difference RNAi ctrl minus Driver ctrl: −1.93 [95.0% CI −4.51, 1.13] *p* = 0.228; *CG18549* knockdown minus Driver ctrl: is −0.14 [95.0% CI −2.69, 2.43] *p* = 0.941; *CG18549* knockdown minus RNAi ctrl: 1.79 [95.0% CI −1.25, 4.86] *p* = 0.311) and *Vmat* (Kruskal–Wallis * *p* = 0.0351; Driver ctrl vs RNAi ctrl MW * *p* = 0.0357, Driver ctrl vs. *CG18549* knockdown MW * *p* = 0.0476; unpaired mean difference RNAi ctrl minus Driver ctrl: −6.01 [95.0% CI −10.5, −2.67] *p* = 0.005; *CG18549* knockdown minus Driver ctrl: −3.97 [95.0% CI −6.27, −0.931] *p* = 0.0188; *CG18549* knockdown minus RNAi ctrl: 2.04 [95.0% CI −0.0492, 7.1] *p* = 0.188).

**Figure 8 insects-12-01024-f008:**
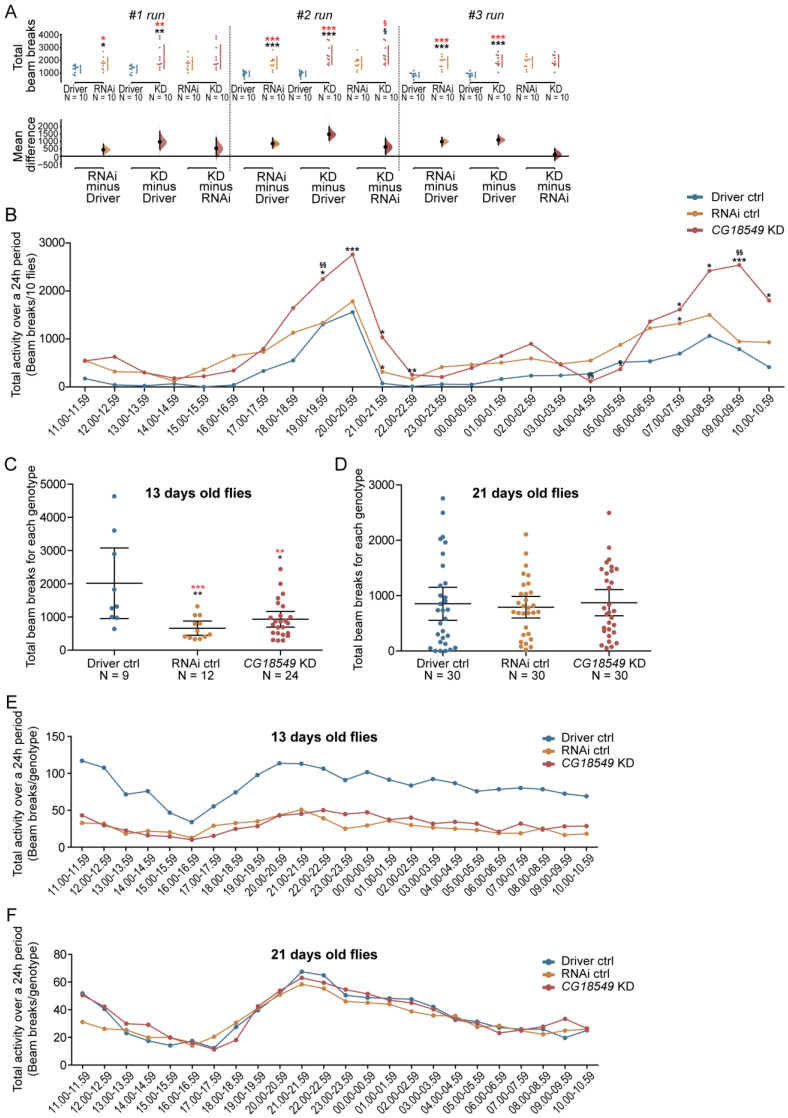
Locomotion on *CG18549* RNAi line 1 using the elav-GAL4 driver. Brain *CG18549* knockdown was performed, and activity was measured in the DAMS system. Five- to seven-days-old male flies (n = 10) were transferred to glass tubes prepared with standard Jazz mix food for 72 h. Differences were calculated using Kruskal–Wallis with Dunn’s comparison and/or Mann–Whitney as a post hoc test to calculate exact *p*-values * or § *p* < 0.0492, ** or §§ *p* < 0.0099, *** or §§§ *p* < 0.0001 (*= significant difference against the Driver control, § = significant difference against the RNAi control, in red effect size differences * = significant difference against the Driver control, § = significant difference against the RNAi control). (**A**) Mean differences for 9 comparisons are shown in the above Cumming estimation plot. The raw data is plotted on the upper axes; each mean difference is plotted on the lower axes as a bootstrap sampling distribution. Mean differences are depicted as dots; 95% confidence intervals are indicated by the ends of the vertical error bars. Kruskal–Wallis Run 1 *p* = 0.0093, Run 2 *p* ≤ 0.0001, Run 3 *p* ≤ 0.0001, Driver ctrl and RNAi Run 1: 4.39 × 10^2^ [95.0% CI 1.14 × 10^2^, 8.14 × 10^2^] *p* = 0.0222, Run 2: 8.52 × 10^2^ [95.0% CI 5.58 × 10^2^, 1.22 × 10^3^] *p* = 0.0, Run 3: 9.81 × 10^2^ [95.0% CI 6.67 × 10^2^, 1.27 × 10^3^] *p* = 0.0; *CG18549* knockdown and Driver ctrl Run 1: 9.69 × 10^2^ [95.0% CI 4.81 × 10^2^, 1.67 × 10^3^] *p* = 0.002, Run 2: 1.48 × 10^3^ [95.0% CI 1.08 × 10^3^, 2.01 × 10^3^] *p* = 0.0, Run 3: 1.1 × 10^3^ [95.0% CI 7.78 × 10^2^, 1.41 × 10^3^] *p* = 0.0; *CG18549* knockdown and RNAi ctrl Run 1: 5.3 × 10^2^ [95.0% CI −12.5, 1.25 × 10^3^] *p* = 0.131, Run 2: 6.28 × 10^2^ [95.0% CI 1.47 × 10^2^, 1.2 × 10^3^] *p* = 0.035, Run 3: 1.19 × 10^2^ [95.0% CI −2.57 × 10^2^, 5.1 × 10^2^] *p* = 0.579. Mann–Whitney post hoc test Run 1: *CG18549* knockdown *p* = 0.0029, RNAi ctrl *p* = 0.0355; Run 2; *CG18549* knockdown * *p* ≤0.0001 § *p* = 0.0312, RNAi ctrl * *p* = 0.0006; Run 3: *CG18549* knockdown ***
*p* ≤ 0.0001, RNAi ctrl * *p* ≤ 0.0001. (**B**) Total activity plotted over time, * *p* < 0.05, ** *p* < 0.001, *** *p* < 0.001 (*= significant difference against the Driver control, § = significant difference against the RNAi control. (**C**–**F**) Brain *CG18549* knockdown was performed, and activity was measured in the DAMS system. Male flies were aged to 13 (Driver ctrl n = 9, RNAi ctrl n = 12, CG18549 knockdown n = 24) and 21 (n = 30) days. They were transferred to glass tubes prepared with standard Jazz mix food for 72 h. Differences were calculated using Kruskal–Wallis with Dunn’s comparison and/or Mann–Whitney as a post hoc test to calculate exact *p*-values * *p* < 0.0492, ** *p* < 0.0099, *** *p* < 0.0001 (* = significant difference against the Driver control, § = significant difference against the RNAi control. (**C**,**D**) Scatter plot displaying the total beam breaks, (**C**) 13 days old flies (Kruskal–Wallis *p* = 0.0063, Mann–Whitney *CG18549* knockdown * *p* = 0.0129, RNAi ctrl * *p* = 0.0040; unpaired mean difference between Driver ctrl and RNAi ctrl is −1.35 × 10^3^ [95.0% CI −2.43 × 10^3^, −6.14 × 10^2^] *p* = 0.0008, Driver ctrl and *CG18549* knockdown is −1.08 × 10^3^ [95.0% CI −2.23 × 10^3^, −3.91 × 10^2^] *p* = 0.0052, RNAi ctrl and *CG18549* knockdown is 2.69 × 10^2^ [95.0% CI −20.0, 5.5 × 10^2^] *p* = 0.131) and (**D**) 21 days old flies (no differences). (**E**,**F**) Total activity plotted over time, (**E**) 13 days old flies and (**F**) 21 days old flies.

**Figure 9 insects-12-01024-f009:**
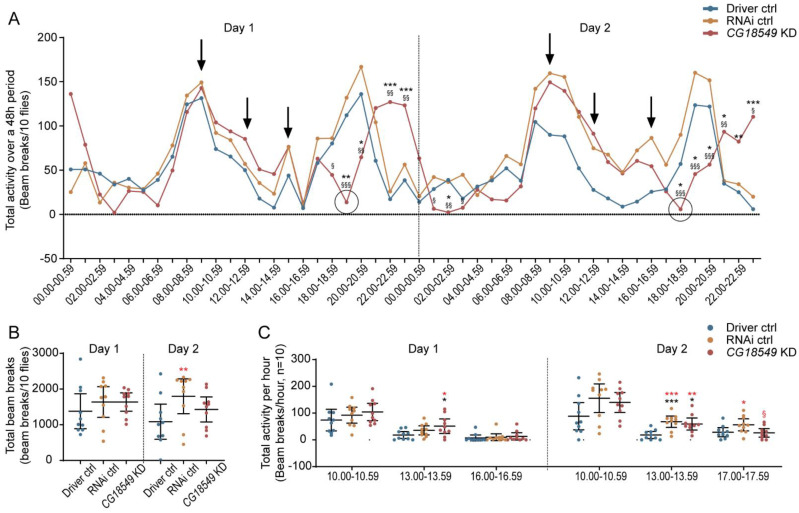
Brain knockdown of *CG18549* affects the stress response after stimulation with bright light. Brain *CG18549* knockdown using the elav-GAL4 driver and the *CG18549* RNAi line 1 was performed, and flies (n = 10) were placed in a DAMS for 48 h). Stress was induced by exposing the flies to bright light (LED, 1500 lumens) for 10 min three times (9.00, 12.00 and 15.00/16.00) per day, Differences were calculated using Kruskal–Wallis with Dunn’s comparison and/or Mann–Whitney as a post hoc test to calculate exact *p*-values * or § *p* < 0.0492, ** or §§ *p* < 0.0099, *** or §§§ *p* < 0.0001 (* = significant difference against the Driver control, § = significant difference against the RNAi control, in red effect size differences * = significant difference against the Driver control, § = significant difference against the RNAi control). (**A**) Total activity over the 48 h period. Black arrows illustrate the stimulation, and the two circulated points show the large dip in activity that the *CG18549* knockdown flies experienced post-stimulation. (**B**) Total beam breaks per genotype divided into day 1 (Kruskal–Wallis *p* = 0.3682, unpaired mean difference between Driver ctrl and RNA ctrl 2.56 × 10^2^ [95.0% CI −3.51 × 10^2^, 7.15 × 10^2^] *p* = 0.379, Driver ctrl and *CG18549* knockdown 2.57 × 10^2^ [95.0% CI −2.8 × 10^2^, 6.53 × 10^2^] *p* = 0.301, RNAi ctrl and *CG18549* knockdown 0.8 [95.0% CI −3.76 × 10^2^, 4.42 × 10^2^] *p* = 0.997) and day 2 (Kruskal–Wallis *p* = 0.0599, unpaired mean difference between Driver ctrl and RNA ctrl 7.11 × 10^2^ [95.0% CI 69.2, 1.22 × 10^3^] *p* = 0.0384, Driver ctrl and *CG18549* knockdown 3.43 × 10^2^ [95.0% CI −1.68 × 10^2^, 8.17 × 10^2^] *p* = 0.202, RNAi ctrl and *CG18549* knockdown −3.68 × 10^2^ [95.0% CI −8 × 10^2^, 1.93 × 10^2^] *p* = 0.169). (**C**) Total activity one-hour post-stimulation [10.00–10.59 day 1: Kruskal–Wallis *p* = 0.2975, effect size RNAi ctrl minus Driver ctrl: 18.3 [95.0% CI −31.5, 53.7] *p* = 0.427, *CG18549* knockdown minus Driver ctrl: 30.1 [95.0% CI −17.7, 70.2] *p* = 0.203, *CG18549* knockdown minus RNAi ctrl: 11.8 [95.0% CI −21.0, 51.2] *p* = 0.548; day 2: Kruskal–Wallis *p* = 0.0849, effect size RNAi ctrl minus Driver ctrl: 67.3 [95.0% CI −0.9, 1.22 × 10^2^] *p* = 0.0576, *CG18549* knockdown minus Driver ctrl: 51.5 [95.0% CI −4.3, 99.3] *p* = 0.077, *CG18549* knockdown minus RNAi ctrl: is −15.8 [95.0% CI −66.2, 41.1] *p* = 0.578; 13.00–13.59 day 1: Kruskal–Wallis *p* = 0.0664, Mann–Whitney * *p* = 0.0444, effect size RNAi ctrl minus Driver ctrl: 17.6 [95.0% CI 0.7, 35.5] *p* = 0.0766, *CG18549* knockdown minus Driver ctrl: 33.1 [95.0% CI 9.9, 59.9]. *p* = 0.0246, *CG18549* knockdown mins RNAi ctrl: 15.5 [95.0% CI −9.6, 43.0] *p* = 0.285; day 2: Kruskal–Wallis *p* = 0.0012, * *p* = 0.0009 and 0.0136, effect size RNAi ctrl minus Driver ctrl: 49.6 [95.0% CI 27.7, 68.0] *p* = 0.0004, *CG18549* knockdown minus Driver ctrl: 41.2 [95.0% CI 21.9, 65.0] *p* = 0.0012, *CG18549* knockdown minus RNAi ctrl: −8.4 [95.0% CI −30.6, 21.2] *p* = 0.53; 16.00–16.59 Kruskal–Wallis *p* = 0.3540, effect size RNAi ctrl minus Driver ctrl: 2.6 [95.0% CI −10.0, 18.1] *p* = 0.64, *CG18549* knockdown minus Driver ctrl: 5.6 [95.0% CI −8.2, 21.0] *p* = 0.469, *CG18549* knockdown minus RNAi ctrl: 3.0 [95.0% CI −12.8, 18.2] *p* = 0.692; 17.00–17.59 Kruskal–Wallis *p* = 0.3682, effect size RNAi ctrl minus Driver ctrl: 27.8 [95.0% CI 4.0, 50.6] *p* = 0.0414, *CG18549* knockdown minus Driver ctrl: −2.4 [95.0% CI −22.7, 15.8] *p* = 0.808, *CG18549* knockdown minus RNAi ctrl: −30.2 [95.0% CI −51.8, −6.9] *p* = 0.0244].

**Table 1 insects-12-01024-t001:** Display the name and the forward and reverse sequence of primers. Housekeeping genes are marked in red.

Primer	Forward	Reverse
* Actin42A *	acaacacttccgctcct	gaacacaatatggtttgcttatgc
*Akh*	tagtgctgtgttaattac	tcattctgagttctatg
*AkhR*	aggagcgactttgatgag	tccgtagcagtagatgaa
*CG18549*	tgttcgtctttacggcattcc	gtgtagccctcacccttgaa
*DAT*	gcttcaaaccataagttctaa	tcggacttgatattatctacaa
*Dilp2*	ctgcagtgaaaagctcaacga	tcattctgagttctatg
*Dilp3*	tgaacgaactatcactcaacagtct	agagaactttggaccccgtgaa
*Dilp5*	gaggcaccttgggcctattc	catgtggtgagattcggagct
*Dilp6*	gtccaaagtcctgctagtcct	tctgttcgtattccgtgggtg
*InR*	caagccgttcgtccatc	tcattccaaagtcaggaa
*Pale*	tcaagaaatcctacagtat	cacaatgcaatcttccag
* Rp49 *	cacaccaaatcttacaaaatgtgtga	aatccggccttgcacatg
* Rpl11 *	ccatcggtatctatggtctgga	catcgtatttctggaacc
*Vmat*	gtgaccttcgggacgatag	actagagcgggaaaaccagc

**Table 2 insects-12-01024-t002:** RNA sequencing log2 fold change as a validation of qPCRs run on the Driver control (*da-GAL4* > *w1118*), the RNAi control (*w1118* > *CG18549 RNAi line 1*) and the CG18549 knockdown (*da-GAL4* > *CG18549 RNAi line 1*) flies. The table summarize the gene, flybase.org (accessed on 16 March 2021) identity (FB ID) and the log2 fold change and Wald test *p*-values.

Gene	FB ID	Wald Test					
		Driver vs. RNAi		Driver vs. Knockdown		RNAi vs. Knockdown	
		Log2 Fold Change	*p*-Value	Log2 Fold Change	*p*-Value	Log2 Fold Change	*p*-Value
*CG18549*	FBgn0038053	0.27	0.57	−0.47	0.32	−0.74	0.15
*Akh*	FBgn0004552	0.35	0.27	0.66	0.04	0.31	0.36
*AkhR*	FBgn0025595	−0.19	0.36	−0.34	0.09	−0.15	0.49
*Ilp2*	FBgn0036046	0.70	0.09	1.16	0.004	0.45	0.30
*Ilp3*	FBgn0044050	0.33	0.34	0.65	0.06	0.32	0.38
*Ilp5*	FBgn0044048	0.09	0.86	1.05	0.04	0.96	0.07
*Ilp6*	FBgn0044047	−0.04	0.76	−0.15	0.31	−0.10	0.50
*InR*	FBgn0283499	N/A	N/A	N/A	N/A	N/A	N/A
*Dat*	FBgn0034136	−0.29	0.27	−0.27	0.30	0.02	0.96
*Dop1R1*	FBgn0011582	−0.56	0.05	−0.56	0.05	−0.01	0.99
*Dop1R2*	FBgn0266137	0.13	0.66	−0.31	0.29	−0.44	0.16
*Dop2R*	FBgn0053517	−0.08	0.90	−1.25	0.03	−1.17	0.06
*Ple*	FBgn0005626	−0.45	0.17	−0.93	0.005	−0.48	0.18
*Vmat*	FBgn0260964	−0.14	0.60	−0.74	0.007	−0.60	0.05

**Table 3 insects-12-01024-t003:** Survival proportions of the Driver control (blue column, *da-GAL4* > *w1118*), the RNAi control (yellow column, *w1118* > *CG18549 RNAi line 1*) and the CG18549 knockdown (red column, *da-GAL4* > *CG18549 RNAi line 1*) flies that were used in the starvation resistance assay. The percentage together with the upper and lower 95% CI for each line at specific hours are presented. A grey row indicates the hours that a line deceased of starvation.

Survival Proportions									
Hours	Driver Control			RNAi Control			*CG18549* Knockdown		
	Percentage	+Error	−Error	Percentage	+Error	−Error	Percentage	+Error	−Error
0.00	100			100			100		
27.36	96.43	3.06	19.18						
27.91				96.67	2.86	18.06			
27.95				93.33	4.96	17.45			
28.00				90.00	6.66	17.88			
28.20				86.67	8.11	18.39			
28.48							96.67	2.86	18.06
28.53				83.33	9.37	18.84			
29.11							93.33	4.96	17.45
29.50							90.00	6.66	17.88
29.75				80.00	10.48	19.20			
30.07				76.67	11.46	19.46			
30.37	92.86	5.31	18.51						
30.53				73.33	12.34	19.64			
31.38							86.67	8.12	18.39
31.53				70.00	13.12	19.74			
31.82	89.29	7.13	18.93						
31.87				66.67	13.81	19.75			
33.23				63.33	14.42	19.69			
33.4				60.00	14.95	19.55			
34.1	85.71	8.67	19.42						
34.13							83.33	9.37	18.84
34.23				56.67	15.41	19.34			
34.98							80	10.48	19.20
35.17							76.67	11.46	19.46
35.18							73.33	12.34	19.64
35.20							70.00	13.12	19.74
35.50				53.33	15.80	19.05			
35.60	82.14	10.01	19.85						
35.73							66.67	13.81	19.75
35.77							63.33	14.42	19.69
35.80							60.00	14.95	19.55
35.82				50	16.12	18.70			
36.87				46.67	16.38	18.27			
36.95				43.33	16.56	17.77			
37.50	78.57	11.18	20.17						
37.67							56.67	15.41	19.34
37.72				40.00	16.67	17.20			
37.92	75.00	12.21	20.40						
38.20	71.43	13.13	20.51						
38.38				36.67	16.71	16.54			
38.43				33.33	16.67	15.81			
38.72				30.00	16.55	14.98			
38.93				26.67	16.35	14.06			
39.02							53.33	15.80	19.05
39.6	67.86	13.94	20.54						
39.87							50.00	16.12	18.70
40.20							46.67	16.38	18.27
40.40	64.29	14.65	20.48						
41.27				23.33	16.05	13.03			
41.62	60.71	15.27	20.32						
42.17							43.33	16.56	17.77
42.40	57.14	15.80	20.09						
42.68				20.00	15.64	11.88			
42.95							40.00	16.67	17.20
43.40				16.67	15.11	10.59			
44.03	53.57	16.25	19.76						
44.98	50.00	16.62	19.36						
45.1	46.43	16.90	18.87						
45.93							36.67	16.71	16.54
46.23	42.86	17.10	18.29						
46.55							33.33	16.67	15.81
46.73				13.33	14.45	9.13			
47.92							30.00	16.55	14.98
48.60	39.29	17.22	17.62						
49.65	35.71	17.26	16.86						
49.82				10.00	13.58	7.45			
50.22	32.14	17.20	16.00						
50.55	28.57	17.04	15.03						
50.75	25.00	16.78	13.94						
50.82				6.67	12.51	5.49			
51.00							26.67	16.35	14.06
51.3							23.33	16.05	13.03
52.18							20.00	15.64	11.88
52.40							16.67	15.11	10.59
52.45							13.33	14.44	9.13
52.73	21.43	16.40	12.72						
55.38	17.86	15.89	11.35						
55.77				3.33	11.18	3.08			
55.8				0	0	0			
56.53							10.00	13.58	7.45
56.75	14.29	15.21	9.79						
57.12	10.71	14.35	7.99						
58.08							6.67	12.51	5.49
62.87							3.33	11.18	3.08
63.67	7.14	13.23	5.89						
64.12	3.57	11.84	3.31						
68.28							0	0	0
74.02	0	0	0						

## Data Availability

The datasets generated for this study can be found in the SRA (SUB8801412, https://submit.ncbi.nlm.nih.gov/subs/sra/SUB8801412/overview, accessed on 16 March 2021) or within the manuscript and its Appendix A.

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
