# Peer review of "The Fly Homologue of MFSD11 Is Possibly Linked to Nutrient Homeostasis and Has a Potential Role in Locomotion: A First Characterization of the Atypical Solute Carrier CG18549 in Drosophila Melanogaster"

_insects, 2021, doi:10.3390/insects12111024_

Round 1

Reviewer 1 Report

The work by Ceder and colleagues reports first attempts to study the function of CG18549, a Drosophila gene encoding an orphan solute carrier (SLC). Elucidating the functional repertoire of transporters is an important question for both basic knowledge and biomedical issues, and in particular the activity of many SLCs remains unknown.

Having shown substantial evidence that CG18549 is homologous to human MFSD11, the authors performed various assays to analyze its putative activity in the regulation of gene expression, response to diet changes and locomotor behavior. Although each of the assays seem to have been carefully executed on a technical point of view, the manuscript in its current form unfortunately displays major caveats.

Major points:

  • The author used an enhancer trap line, called CG18549-Gal4, as means to infer the expression pattern of the CG18549. Firm evidence that this driver captures at least in part the endogenous expression of CG18549 yet remains to be established. For example, CG18549 displays prominent and specific expression in the embryonic tracheal system, as reported in Flybase and previously published by Chung et al. Dev Biol 2011 (PMID: 21963537). Does the Gal4 line drive a similar pattern? Reciprocally, I wonder whether the Gal4 pattern seen in larval and adult brains could also be detected by in situ hybridization? In the adult, CG18549 is strongly expressed in the excretory Malpighian tubules, the insect renal system, which can make good sense for a solute carrier. Is the Gal4 line driving expression in this tissue?

The most problematic issue further concerns the lack of factual evidence that the observed phenotypes can be unambiguously attributed to the loss of CG18549 function. In the absence of genetic mutant, the authors made use of two transgenic RNAi lines targeting CG18549, and RNAi approaches are prone to artefacts (e.g., off-target effects). The system is based on two separate components, an UAS-RNAi line and a selected Gal4 line. Following their cross, only the progeny that contains both constructs can drive RNAi expression and thus knockdown the gene of interest. However, in many instances either the Gal4-driver or the RNAi parental lines lead to defects, with comparable amplitude or statistical significance when compared to CG18549 KD conditions (Gal4 driver+UAS-RNAi).

  • This critical problem is well illustrated in Figure 3, where RNAi line #2 alone is clearly different from the driver line alone, while RNAi line #2 is not different from KD conditions (p-val >0.5, Figure 3B), thus preventing any conclusions regarding a specific effect of CG18549 The same is also true in Figure 5C, when assaying the expression of Akh, or in Figure 6E for quantification of Pale and Vmat levels.

Therefore, the use of RNA line #2 is not appropriate and should not been considered as a means to inactivate CG18549.

  • Unfortunately, the use of RNA line #1 is also problematic. For example, locomotion monitoring assays detected a significant difference in total beam breaks between individual driver and RNAi lines (Figure 6A), but no difference between the driver and KD (p-val >0.6). In the measurement of total activity over a 24H period, the driver appears very similar to KD conditions, both seemingly different from the RNAi alone (Figure 6B).
  • Similar concerns apply to Figure 7, in which the effects seen for RNAi alone are close to those seen upon CG18549 KD, when exploring locomotion (Figure 7A,B) or gene expression (Figure 7C). Again, the same is true for Figure 8, as seen for different runs (Figure 8A), or for 13 days old flies (Figure 8C and E), or even for Figure 9 B,C where CG18549 knockdown conditions seem indistinguishable from RNAi alone.
  • Finally, comparing RNA-seq data between the driver line, RNAi line, and KD conditions should provide a direct estimate of the efficacy of CG18549 From a very quick look at the supplemental data, I didn’t detect significant changes in CG18549 levels in knockdown conditions. Could the authors further elaborate on this specific point? Is there any evidence of reduced levels of CG18549 in RNA-seq analysis of KD animals?

In sum, given the serious limitations of these RNA lines, I’m afraid that unambiguous conclusions on the putative function of CG18549 cannot be attained based on current data.

Minor points:

  • In which tissues and/or stages the effects of CG18549 KD on gene expression (qPCR and RNA-seq) have been evaluated remains not clear for me.
  • The authors report that changes in the sugar/protein balance affect positively or negatively CG18549 RNA levels. Is Gal4 line activity also sensitive to diet?
  • Throughout the manuscript, the authors use multiple pairwise tests comparing: driver vs RNAi, driver vs KD and RNAi vs KD. To compare three or more samples, pairwise tests are generally inappropriate and other methods should be used (e.g., one-way ANOVA or Kruskal-Wallis non-parametric tests if values do not follow a gaussian distribution). Why did the authors choose these analyses? Indeed, they mentioned Kruskal-Wallis tests but, if I understood properly, used Mann-Whitney pairwise tests to calculate p-values. I am curious to see p-values calculated by Kruskal-Wallis tests, e.g. in Fig3B.

Author Response

First of all thank you for your comments and concerns, we have answered them in the attached document, please see the attachment.

Reviewer 2 Report

The authors investigated the drosophila homolog of MFSD11. While their results are technically sound, the presentation of the results needs to be improved. Lists of statistical parameters , for instance, should be moved from the result section into the figure legends. In the result section on the other hand the rational behind the conducted experiments ought to better explained. 

SLCs are indeed understudied and knowing what this transporter does is important.
The insights provided by the present study are minimal.
It is unclear why the authors chose to conduct the described experiments. No clear rational is given.
There are hardly any conclusions. The discussion is riddled with speculations.
The reference are fine.
Statistical parameters should not be listed in the result section. They should either be presented in tables or listed in the figure legends. 

Author Response

First of all, thank you so much for your comments and questions, please see the attachment for answers. 

Reviewer 3 Report

The article  is very well written and brings new informations in the fild.

I would like to congratulate the authors for their hard work

A few comments regarding the article
-please detailed the full name for mTORC1-line 45

The article brings new information about  MFSD11, but  MFSD11 has been deeply characterized in other species? If so, please specify  

How many  studies support the idea  that  CG18549 has a potential role in locomotion behavior and  metabolism in pro and eukaryotes?

Author Response

Thank you so much for your comments, please see the attachment for answers.

Round 2

Reviewer 1 Report

The authors did not perform any additional experiments and basically respond to my comments through substantial changes in the text, toning down several of their conclusions, including the title. They have also modified the manuscript according to suggestions of one other  reviewer.

Reviewer 2 Report

The authors have improved the original draft. I therefore recommend acceptance.